

# The importance of comprehensive parameter sampling and multiple observations for robust constraint of aerosol radiative forcing

Jill S. Johnson[1], Leighton A. Regayre[1], Masaru Yoshioka[1], Kirsty J. Pringle[1], Lindsay A. Lee[1], David Sexton[2], John Rostron[2], Ben B. B. Booth[2] and Ken S. Carslaw[1]

[1]School of Earth and Environment, University of Leeds
[2]Met Office, Exeter, UK

*Correspondence to*: Jill S. Johnson (j.s.johnson@leeds.ac.uk)

**Abstract.** Observational constraint of simulated aerosol and cloud properties is an essential part of building trustworthy climate models for calculating aerosol radiative forcing. Models are usually tuned to achieve good agreement with observations, but tuning produces just one of many potential variants of a model, so the model uncertainty cannot be determined. Here we estimate the uncertainty in aerosol effective radiative forcing (ERF) in a tuned climate model by constraining 4 million variants of the HadGEM3-UKCA aerosol-climate model to match nine common observations (top-of-atmosphere shortwave flux, aerosol optical depth, PM$_{2.5}$, cloud condensation nuclei, concentrations of sulphate, black carbon and organic carbon, as well as decadal trends in aerosol optical depth and surface shortwave radiation.) The model uncertainty is calculated by using a perturbed parameter ensemble that samples twenty-seven uncertainties in both the aerosol model and the physical climate model. Focusing over Europe, we show that the aerosol ERF uncertainty can be reduced by about 30% by constraining it to the nine observations, demonstrating that producing climate models with an observationally plausible "base state" can contribute to narrowing the uncertainty in aerosol ERF. However, the uncertainty in the aerosol ERF after observational constraint is large compared to the typical spread of a multi-model ensemble. Our results therefore raise questions about whether the underlying multi-model uncertainty would be larger if similar approaches as adopted here were applied more widely. It is hoped that aerosol ERF uncertainty can be further reduced by introducing process-related constraints, however, any such results will be robust only if the enormous number of potential model variants is explored.

## 1   Introduction

It has proven extremely challenging to reduce the large uncertainty in aerosol model simulations and the calculated aerosol radiative forcing since pre-industrial times. Although extensive improvements in the physical realism of aerosol-climate models have been made in recent years (Ghan and Schwartz, 2007; Mann et al., 2014), aerosol model simulations are still surprisingly uncertain – up to a factor ten or more spread in key aerosol properties among models (Mann et al., 2014). Calculated aerosol radiative forcing also remains stubbornly uncertain among multiple models (Boucher et al., 2013; Myhre





et al., 2013), although the set of models is different to those used to assess aerosol microphysical properties in Mann et al. (2014), making it difficult to establish the causes of forcing uncertainty. Until the uncertainty is reduced, climate models will not be robust in their predictions of decadal-scale climate change and its global and regional impacts (Andreae et al., 2005; Myhre et al., 2013; Seinfeld et al., 2016).

The uncertainty in aerosol radiative forcing has persisted through multiple generations of climate models because it results from the combined effects of dozens of complex and uncertain climate model processes related to aerosols, clouds, radiation and dynamics. Changes in aerosols cause the entire aerosol-cloud-radiation-dynamics system to respond, resulting in an Effective Radiative Forcing, or ERF (Boucher et al., 2013). The complexity of the processes causing the aerosol ERF (and the

fact that it cannot be measured directly) means that it may essentially be treated as a tuneable model quantity (Hourdin et al., 2016; Mauritsen et al., 2012) rather than being properly constrained by extensive measurements that define the state and behaviour of aerosols and clouds. This is not a firm basis for climate projections.

There are three ways in which observations help to constrain the uncertainty in aerosol ERF. The first is based on the

recognition that the forcing depends on the interlinked sensitivities of aerosols, clouds and their radiative properties to changes in aerosol emissions. For example, the magnitude of the aerosol-cloud interaction component of radiative forcing ($R$) can be broken down into a product of sensitivities relating the forcing to aerosol emissions ($E$), cloud condensation nuclei concentrations ($N_{CCN}$) and droplet concentrations ($N_d$) (Ghan et al., 2016):

$$\frac{d \ln R}{d \ln E} = \frac{d \ln N_{CCN}}{d \ln E} \times \frac{d \ln N_d}{d \ln N_{CCN}} \times \frac{d \ln R}{d \ln N_d}$$

Relationships between various aerosol, cloud and radiation variables are widely used or proposed as a way of constraining the uncertainty in aerosol-cloud forcing in climate models (Ban-weiss et al., 2014; Grandey et al., 2013; Gryspeerdt et al., 2016, 2017b; Gryspeerdt and Stier, 2012; Lebo and Feingold, 2014; McCoy et al., 2016; Quaas et al., 2009, 2010; Terai et al., 2015;

Yi et al., 2012; Zhang et al., 2016).

The second aspect of model constraint is to test the model's ability to reproduce observed trends in aerosols, clouds and radiation (Allen et al., 2013; Cherian et al., 2014; Leibensperger et al., 2012; Li et al., 2013; Liepert and Tegen, 2002; Shindell et al., 2013; Turnock et al., 2015; Zhang et al., 2017). For example, Cherian et al. (2014) showed that among several climate

models there is a relationship between the simulated trend in European surface solar radiation (SSR) over recent decades and the pre-industrial to present-day aerosol ERF (models with large SSR trends tend to simulate larger ERFs). Cherian et al.



(2014) used this relationship to define the observationally constrained ERF based on the models that simulate SSR trends closest to observations (a so-called emergent constraint).

The third aspect of model constraint is to observationally constrain the model "base state" – i.e., properties like aerosol optical
depth (AOD) or aerosol concentrations in a particular period. Considerable effort is put into constraining the model base state because observations are readily available and models that cannot simulate aerosol and cloud properties close to observations would not be trusted to predict changes in these properties over time (which determines the forcing). Models can also be constrained under a range of cloud regimes as well as under pristine and polluted conditions, which will have a bearing on a model's ability to simulate the change from the pre-industrial period to the present-day (Carslaw et al., 2013, 2017). Model
skill in simulating AOD was used in the Atmospheric Chemistry-Climate Model Intercomparison Project to screen the models (Shindell et al., 2013) and global AOD reanalysis products have been used to help constrain the aerosol forcing (Bellouin et al., 2013). It is also argued that the wealth of available measurements will help to constrain direct radiative forcing (Kahn, 2012)

There are limitations with all three methods outlined above in terms of constraining the uncertainty in aerosol forcing over periods outside the observational record. The main limitation with the first method (aerosol-cloud-radiation relations) is that there is no guarantee that present-day (or "instantaneous") relationships can be extrapolated to pristine pre-industrial conditions (Penner et al., 2011). Even the most sophisticated approaches still rely on model estimates of how aerosols changed over the industrial period (Gryspeerdt et al., 2017a). The same main limitation applies to the second method (aerosol and radiation
trends): most data records are quite short so typically do not include pristine pre-industrial-like conditions (Carslaw et al., 2017; Hamilton et al., 2014). With the third method (constraining the state of aerosols, clouds and their radiative properties) it is not obvious how the model accuracy can be related to the uncertainty in simulated radiative forcing (i.e., there is no equivalent to Equation 1 defining how a bias in simulated aerosol properties affects the forcing). One aim of our study is therefore to make that link.

In this paper we focus on observationally constraining uncertainty in the base state of an aerosol-climate model as well as trends in radiative properties. Our approach is shown schematically in Figure 1. We begin with a large set of model variants produced by adjusting multiple uncertain model input parameters (a tiny fraction of which would be explored in model tuning). These model variants (parameter combinations) define the prior model uncertainty (which can be defined by a pdf), which we
then constrain by identifying variants that produce plausible outputs compared to aerosol and cloud observations. Model variants that produce implausible results are rejected and, likewise, the forcings that they calculate are also rejected. A similar constraint methodology has been applied to environmental models (Salter and Williamson, 2016), hydrological models (Liu and Gupta, 2007), galaxy formation (Rodrigues et al., 2017), disease transmission (Andrianakis et al., 2017), climate models (Murphy et al., 2004; Regayre et al., 2018; Sexton et al., 2011; Williamson et al., 2013)and aerosol models (Lee et al., 2011,





2013; Reddington et al., 2017; Regayre et al., 2018, 2014, 2015). In this paper the observations comprise aerosol and cloud state variables and trends, but the approach could readily be extended to include any observations, such as of aerosol-cloud-radiation relationships.

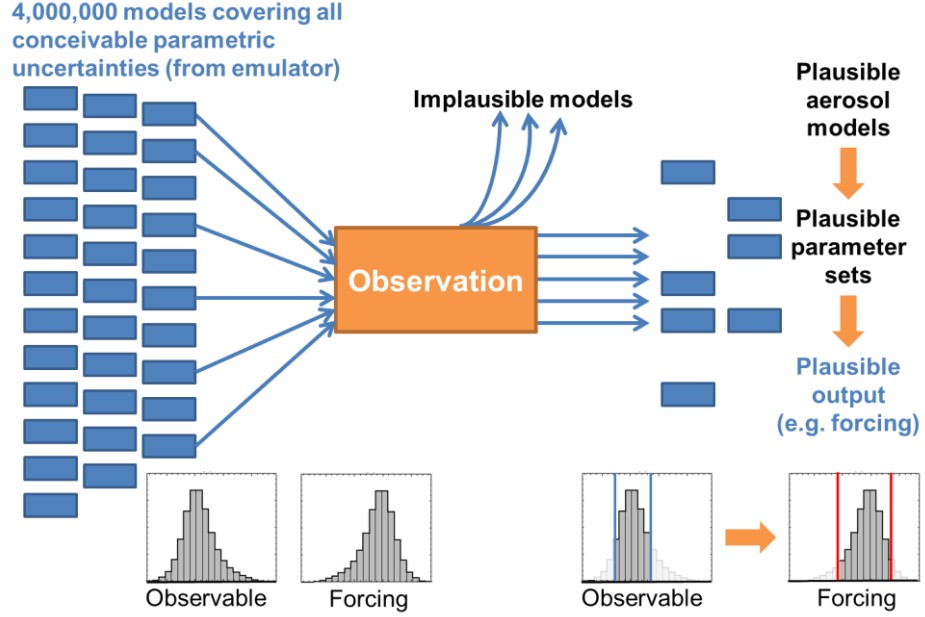

**Figure 1.** Schematic of the methodology for observational constraint of parametric model uncertainty.

We define observational constraint as *finding the full set of model variants that can be considered plausible against observations*, and from which we can estimate the prior (unconstrained) and remaining (observationally constrained) uncertainty. This approach is different to traditional model tuning, which produces only one result on the right side of Fig 1 with no information about uncertainty. We note, however, that such model adjustments towards observations are often misleadingly called constraint.

15 The vast majority of observational constraint efforts are severely limited by the very small number of models used, which makes it impossible to reach robust statistical conclusions about model uncertainty. In a multi-model ensemble the number of models is often about ten or so, and in model tuning perhaps only a few dozen parts of parameter space are explored. To get around this problem we build emulators that enable model outputs to be generated for millions of model parameter combinations (Lee et al., 2011, 2013), which enables us to relate the uncertainty on the left side of Fig. 1 (in the form of a pdf)

20 to the observationally constrained uncertainty on the right side.



The main aims of this paper are first to determine how much uncertainty could potentially remain in an aerosol-chemistry-climate model that is tuned to match various sets of observations, and second, how this uncertainty might affect conclusions drawn from multi-model ensemble studies which do not explicitly account for this source of uncertainty. Although large observational datasets of aerosol in-situ microphysical and chemical properties are available (Reddington et al., 2017), we use synthetic observations here to postpone addressing some of the challenges faced when comparing model output and in-situ observations (Schutgens et al., 2016a, 2016b).

The analysis is restricted to Europe for the month of July. We do this primarily because regional observations can provide a better constraint on model uncertainty than global mean observations (Regayre et al., 2018). The sources of uncertainty in aerosols and forcing vary regionally (Lee et al., 2016; Reddington et al., 2017; Regayre et al., 2015), so a global analysis would essentially be a scaled-up version of what we present here, but with the disadvantage of being less straightforward to understand. We choose Europe because there are many long-term measurements of different aerosol and radiative properties available across it, which we can use to inform our assessments of observational uncertainty.

The following section describes the aerosol-climate model, the set-up of the simulations and the statistical methodology we use to sample the model uncertainty. Section 3 describes our results, starting with an analysis of the magnitude and causes of model uncertainty. We then examine the effects of observational constraint on the simulated aerosol properties, the multi-century (1850-2008) and multi-decade (1978-2008) aerosol ERF uncertainty and the plausible parameter ranges. In section 4 we estimate the potential implications of our results for multi-model emergent constraint studies and other studies that use observations to screen out models.

## 2  Methods

### 2.1  Summary of the constraint methodology

The steps involved are (Figure 1):

1. A perturbed parameter ensemble (PPE) of the HadGEM3-UKCA aerosol-chemistry-climate model (section 2.2) is created to efficiently sample combinations of 27 uncertainties related to the aerosol model and physical processes in the host climate model (mostly related to clouds). The PPE (section 2.3) consists of three sets of 191 single-year simulations which differ only in the anthropogenic aerosol emissions prescribed (1850, 1978 and 2008). The use of HadGEM enables us to diagnose the aerosol ERF rather than just the cloud albedo forcing as in our previous studies (Carslaw et al., 2013; Regayre et al., 2014, 2015).





2. Emulators are built based on the PPE training data (step 1) which define (within quantifiable uncertainty) how aerosol properties and aerosol radiative forcing vary over the 27-dimensional parameter space (section 2.4). We validate each emulator's ability to reproduce model output, then use them to sample the 4 million Monte Carlo points from the parameter space to produce the set of model variants on the left side of Fig 1. This step is essential because, with 27 dimensions of model

uncertainty, the 191 PPE simulations are sparsely distributed. A denser sample of the multi-dimensional parameter space from the emulator enables us to conduct robust statistical analyses.

3. The causes of uncertainty in the aerosol and forcing variables are determined using variance-based sensitivity analysis (section 2.5). This step is not essential for constraining the model, but is useful for understanding which processes in the model

account for the uncertainties in the outputs (Carslaw et al., 2013; Lee et al., 2013; Regayre et al., 2018, 2014, 2015).

4. A set of 'synthetic' observations (section 2.6) is created with realistic uncertainty ranges. We use one PPE member to define these synthetic observations.

5. We identify which of the 4 million model variants are consistent with the observations within their individual uncertainty ranges (section 2.7). This reduced set of variants defines the ways in which parameter values can be combined to reproduce multiple observations and is equivalent to identifying several thousand equally plausible tuned HadGEM3-UKCA models. This procedure is often called 'history matching' or 'pre-calibration' (Craig et al., 1997; Edwards et al., 2011; Williamson et al., 2013; Lee et al., 2016; Andrianakis et al., 2017).

6. The reduction in aerosol ERF uncertainty is quantified using the observationally constrained parameter space (section 2.8).

The observational constraint approach we apply here is quite different to aerosol data assimilation (Bellouin et al., 2013), which cannot directly estimate aerosol ERFs nor the uncertainty. In principle, both approaches should generate similar

distributions of AOD (the usual assimilated observation variable) if similar observations are used. However, we can directly determine aerosol ERF and its uncertainty by running the plausible model variants in both the present-day (where the model uncertainty was constrained) and the pre-industrial. In contrast, estimation of the ERF using the assimilation approach relies on assumptions about how present-day natural AOD represents pre-industrial aerosols because the approach generates only a present-day aerosol state and not a model that can be used to simulate pre-industrial conditions.

## 2.2  The HadGEM3-UKCA climate model

We use the UK Hadley Centre Unified Model HadGEM3 (HadGEM3, 2017) incorporating version 8.4 of the UK Chemistry and Aerosol (UKCA) model. UKCA simulates trace gas chemistry and the evolution of the aerosol particle size distribution



and chemical composition using the GLObal Model of Aerosol Processes (GLOMAP-mode; (Mann et al., 2010)) and a whole-atmosphere chemistry scheme (O'Connor et al., 2014). The model has a horizontal resolution of 1.25x1.875 degrees and 85 vertical hybrid pressure levels.

The aerosol size distribution is defined by seven log-normal modes: one soluble nucleation mode as well as soluble and insoluble Aitken, accumulation and coarse modes. The aerosol chemical components are sulphate, sea salt, black carbon (BC), organic carbon (OC) and dust. Secondary organic aerosol (SOA) material is produced from the first stage oxidation products of biogenic monoterpenes under the assumption of zero vapour pressure. SOA is combined with primary particulate organic matter after kinetic condensation.

GLOMAP simulates new particle formation, coagulation, gas-to-particle transfer, cloud processing and deposition of gases and aerosols. The activation of aerosols into cloud droplets is calculated using globally prescribed distributions of sub-grid vertical velocities (West et al., 2014) and the removal of cloud droplets by autoconversion to rain is calculated by the host model. Aerosols are also removed by impaction scavenging of falling raindrops according to the parametrisation of clouds and
precipitation collocation (Boutle et al., 2014; Lebsock et al., 2013). Aerosol water uptake efficiency is determined by kappa-Kohler theory (Petters and Kreidenweis, 2007) using composition-dependent hygroscopicity factors.

Anthropogenic emission scenarios prepared for the Atmospheric Chemistry and Climate Model Inter-comparison Project (ACCMIP) and prescribed in some of the CMIP Phase 5 experiments are used here. Carbonaceous aerosol emissions for recent
decades were prescribed using a ten year average of 2002 to 2011 monthly mean data from the Global Fire and Emissions Database (GFED3; (van der Werf et al., 2010)) and according to Lamarque et al. (2010) for 1850. The prescribed volcanic $SO_2$ emissions combine emissions from the Andres and Kasgnoc (1998) dataset for continuously erupting and sporadically erupting volcanoes and the Halmer et al. (2002) dataset for explosive volcanoes.

Horizontal winds in the simulations are nudged towards European Centre for Medium-Range Weather Forecasts (ECMWF) ERA-Interim reanalyses for 2006 between approximately 2.15 and 80 km using a 6-hour relaxation timescale. Nudging means that pairs of simulations have near-identical synoptic-scale features, which enables the effects of perturbations to aerosol and chemical processes within the boundary layer to be quantified using single-year simulations. Without nudging, the model fields would need to be averaged over several decades in order to produce signals stronger than the noise caused by internal variability
(Kooperman et al., 2012). By nudging horizontal winds but not temperature, liquid water path and atmospheric humidity can respond to aerosol-induced changes in temperature, allowing more of the rapid responses of clouds and radiation to aerosol perturbations to be captured.



Each simulation was subject to a four month spin-up period with parameters set to their median values. Parameter perturbations were then applied distinctly to individual ensemble members and spun-up for a further 9 months. We analyse the data from July for each simulation following the spin-up period. The calculation of the aerosol ERF and its components is described in section 2.8.

## 2.3 Perturbed parameter ensemble

A perturbed parameter ensemble (PPE) is a set of simulations with excellent space-filling properties that provides information about model output across the multi-dimensional space of uncertain model input parameters. The PPE, described in detail in Yoshioka et al. (2018), was specifically designed to sample aerosol as well as host physical climate model parameters of

importance to the aerosol ERF. Regayre et al. (2018) show that host model parameters cause most of the uncertainty in the radiative state of the atmosphere but aerosol parameters contribute more to the uncertainty in the change-of-state uncertainty (aerosol ERF) .

The 27 perturbed parameters are listed in Table 1. They are categorized as either aerosol (aer) or atmospheric (atm) according

to their role in the model. To define the set of parameters we used expert elicitation and carried out one-at-a-time parameter perturbation screening experiments to quantify the effect of individual parameter perturbations away from the default setting.

Eighteen parameters related to aerosol and precursor gas emissions, deposition and aerosol processes were perturbed based on their importance as causes of uncertainty in aerosols and aerosol-cloud forcing (Lee et al., 2013, Carslaw et al., 2013, Regayre

et al., 2014, 2015). Several parameters available in the HadGEM3-UKCA model but not in the chemistry transport model were included after analyzing the one-at-a-time perturbation screening experiments. These are the updraft velocity in shallow clouds, the fraction of large-scale cloud in which rain-scavenging of aerosols can occur, and the refractive indices of BC and OC. In some cases we perturbed similar parameters as in Regayre et al. (2014) but these parameters are handled differently within the HadGEM model. These are the dry deposition velocity of $SO_2$, dust emissions, and the fraction of ice in mixed-phase clouds

above which aerosol scavenging is suppressed. These parameters are described in more detail in related papers (Regayre et al., 2018; Yoshioka et al., 2018).

Nine physical model parameters were perturbed. These were selected from a much larger set tested by the UK Met Office in developing their ensemble prediction system (Sexton et al., 2017) based on their potential to contribute to uncertainty in a

broad range of aerosol, cloud and radiation properties; in particular particle number concentrations, cloud condensation nuclei, $PM_{2.5}$, aerosol optical depth, sulphate and SOA concentrations, cloud reflectivity, liquid water path, precipitation and aerosol ERF. These 9 atmospheric model parameters are considered the most likely causes of uncertainty in low-altitude clouds and




aerosol-cloud interactions because they influence boundary layer clouds by altering cloud radiative properties, cloud drop concentrations and microphysical processes, atmospheric humidity, convection processes and boundary layer stability.

A probability density function was defined for each parameter to represent shared expert beliefs about parameter uncertainty.

These distributions have no effect on the model simulations (although the ranges define the span of the parameter space), but are used at the stage of generating probability distribution functions (pdfs) of model output based on Monte Carlo sampling from the emulators. We used mainly trapezoidal distributions that avoid overly-centralised Monte-Carlo sampling of the multi-dimensional parameter space (Yoshioka et al., 2018).

Maximin Latin Hypercube sampling was used to create an initial set of 162 simulations that sample model output across the 27-dimensional parameter space. A further set of 54 simulations was used to validate the emulators. In total 217 perturbed parameter simulations were run for each anthropogenic emission period (1850, 1978 and 2008 emissions). Twenty-five simulations did not complete an annual cycle so the ensemble of simulations for each period was therefore made up of the remaining 191 simulations, all of which were used to build the final emulators. Radiative forcings were calculated as the

difference in top-of-atmosphere (ToA) radiative flux for pairs of simulations with identical parameter settings but different anthropogenic emissions (1850, 1978 and 2008).

| Index | Name | Type | Description |
|---|---|---|---|
| 1 | Rad_Mcica_Sigma | Atm | Fractional standard deviation of sub-grid condensate seen by radiation (controls the overlap of sub-grid clouds) |
| 2 | C_R_Correl | Atm | Cloud and rain sub-grid horizontal spatial correlation (determines the accretion rate of cloud drops and aerosols by rain drops) |
| 3 | Niter_Bs | Atm | Number of microphysics iteration sub-steps |
| 4 | Ent_Fac_Dp | Atm | Entrainment amplitude scale factor (controls the convective mass flux and sensitivity of convection to relative humidity) |
| 5 | Amdet_Fac | Atm | Mixing detrainment rate scale factor (controls the rate of humidification of the atmosphere and the shape of the convective heating profile) |
| 6 | Dbsdtbs_Turb_0 | Atm | Cloud erosion rate per second (The rate at which unresolved sub-grid motions mix clear and cloudy air) |
| 7 | Mparwtr | Atm | Maximum value of the function controlling convective parcel maximum condensate |
| 8 | Dec_Thres_Cld | Atm | Threshold for cloudy boundary layer decoupling |
| 9 | Fac_Qsat | Atm | Rate of change in convective parcel maximum condensate |
| 10 | Ageing | Aer | Ageing of hydrophobic aerosols (no of monolayers of soluble material) |
| 11 | Cloud_pH | Aer | pH of cloud droplets (used to calculate the conversion of $SO_2$ into sulphate) |
| 12 | Carb_BB_Ems | Aer | Carbonaceous biomass burning emissions scale factor |
| 13 | Carb_BB_Diam | Aer | Carbonaceous biomass burning emission diameter (nm) |
| 14 | Sea_Spray | Aer | Sea spray aerosol scale factor |
| 15 | Anth_SO2 | Aer | Anthropogenic $SO_2$ emission scale factor |
| 16 | Volc_SO2 | Aer | Volcanic $SO_2$ emission scale factor |




| 17 | BVOC_SOA | Aer | Biogenic secondary aerosol formation from volatile organic compounds scale factor |
| 18 | DMS | Aer | Dimethyl sulphide surface ocean concentration scale factor |
| 19 | Dry_Dep_Acc | Aer | Accumulation mode dry deposition velocity scale factor |
| 20 | Dry_Dep_SO2 | Aer | $SO_2$ dry deposition velocity scale factor |
| 21 | Kappa_OC | Aer | Kappa-Kohler coefficient of organic carbon |
| 22 | Sig_W | Aer | Updraft vertical velocity standard deviation (used to calculate the activation of aerosols into cloud drops) |
| 23 | Dust | Aer | Dust emission scale factor |
| 24 | Rain_Frac | Aer | Fraction of cloud-covered area in large-scale clouds where aerosol scavenging by rain drops occurs |
| 25 | Cloud-Ice_Thresh | Aer | Threshold of cloud ice fraction above which nucleation scavenging is suppressed (restricting further activation of aerosols into cloud drops) |
| 26 | BC_RI | Aer | Imaginary part of black carbon refractive index |
| 27 | OC_RI | Aer | Imaginary part of organic carbon refractive index |

Table 1: The 27 host model and aerosol parameters included in the PPE. Further details are provided in a separate publication (Regayre et al., 2018).

## 2.4 Model emulation and Monte Carlo sampling

For each model output (such as ToA flux, CCN, etc.) we construct a statistical emulator model over the 27-dimensional parameter uncertainty using the 137 training simulations and validate it using the 54 validation simulations (as described in Lee et al., 2011). Once validated, a further new emulator is then created using both the training and the validation simulations of the PPE, to obtain a final emulator based on all of the information that our simulations contain. We then use this emulator to predict the model output for a large sample (here 4 million) of parameter input combinations that span the 27-dimensional parameter space of the PPE. From this sample we obtain a pdf of the uncertainty in this output variable caused by the defined uncertainty in the model parameters (left hand side of Figure 1). In each case, the output pdf can be sampled according to the elicited parameter probability distributions (Yoshioka et al., 2018), in which case the pdf accounts for prior beliefs about the likelihood of different parameter values. Alternatively uniform sampling can be applied, in which case the output pdf assumes that all parameters have equal likelihood of lying between their elicited upper and lower limits. Our approach is to use the prior probability distributions, informed by expert knowledge, to sample the parameter combinations of the 4 million model variants over the 27-dimensional parameter uncertainty space.

## 2.5 Sensitivity Analysis

Sensitivity analysis (Lee et al., 2011; Saltelli et al., 1999) is used to decompose the uncertainty in European regional mean aerosol properties, trends and forcing variables for July into contributions from each individual model parameter. Here, we use



the extended-FAST method (Saltelli et al., 1999) in the R package 'sensitivity' (Pujol et al., 2008) to sample from the emulators (as described in section 2.4) and decompose the variance into its individual sources. We then calculate the percentage by which the total variance (for a specific model output) would be reduced if the value of the parameter in question was known precisely. These percentage reductions are used in the analysis of the main causes of model uncertainty in section 3.3.

## 2.6 Synthetic observations

The 'observations' are taken from the output of the PPE member with each model parameter set to the median value from its corresponding elicited prior distribution. This PPE member was chosen as it lies reasonably centrally within the 27-dimensional parameter uncertainty space. We also tested a marginal set of observations (from a PPE member that had many

parameter values located towards the edges of their uncertainty range) but the conclusions of our study did not change, so we focus on the results from the more centralised choice of observations.

We use synthetic observations (Table 2) of European July-mean cloud condensation nuclei (CCN) concentration at 0.2% supersaturation at approximate cloud-base height, surface concentrations of $PM_{2.5}$, mass concentrations of sulphate, OC and

BC at the surface, AOD at a wavelength of 550 nm, and the change in AOD ($\Delta$AOD) and surface solar radiation ($\Delta$SSR) between 1978 and 2008. The period 1978 to 2008 was originally chosen because it is an interesting period for global and regional forcing changes. Although AOD measurements are not available back to 1978, this is not vital to the present study which aims to assess potential constraint over a period with substantial aerosol changes. We also constrain the outgoing shortwave radiative flux at the top of the atmosphere (ToA flux) to observations from the Clouds and the Earth's Radiant

Energy System (CERES) with an estimated uncertainty in line with IPCC estimates (Hartmann et al., 2013).

The observation uncertainties are based on our judgement about the combined effect of instrument uncertainties and the uncertainty associated with measurement representativeness (colocation of high-frequency point measurements within low-spatial-resolution, monthly-mean model output subject to meteorological variability (Reddington et al., 2017; Schutgens et al.,

2016a, 2016b). In the constraint process we also account for the emulator error (i.e., the estimated uncertainty in each of the 4 million points associated with using the emulator instead of the model itself).

There are other constraints that could be applied to the model such as the aerosol spatial distribution (Myhre et al., 2009), aerosol vertical profile, absorption AOD and single-scatter albedo. It would also be possible to screen the model variants

according to skill in capturing high temporal resolution variability (Myhre et al., 2009) or skill in different regions dominated by different aerosols (Shindell et al., 2013). Here, in this idealized constraint exercise, we restrict the analysis to monthly mean aerosol properties over Europe.



| Observable Quantity | Value | Uncertainty Range |
|---|---|---|
| Top of atmosphere upward SW flux (W m$^{-2}$) | 129 | 122 – 135 |
| Change in surface downward solar radiation from 1978 to 2008, $\Delta$SSR | 3.8 | 3.2 – 4.4 |
| Cloud condensation nucleus (CCN) conc. at 0.2% supersaturation (cm$^{-3}$) | 536 | 483 – 590 |
| Aerosol optical depth (AOD) | 0.17 | 0.14 – 0.19 |
| Change in AOD from 1978 to 2008, $\Delta$AOD | -0.05 | -0.06 – -0.04 |
| PM$_{2.5}$ mass conc. ($\mu$g m$^{-3}$) | 8.0 | 7.2 – 8.8 |
| Particle sulphate conc. ($\mu$g m$^{-3}$) | 1.7 | 1.2 – 2.2 |
| Particle OC conc. ($\mu$g m$^{-3}$) | 4.4 | 3.9 – 4.8 |
| Particle BC conc. ($\mu$g m$^{-3}$) | 0.23 | 0.21 – 0.26 |

**Table 2:** Observed quantities and corresponding uncertainty ranges used for the constraints applied over Europe. Values are a European July mean.

## 2.7 Identification of plausible model variants

Observationally plausible model variants are defined to be those that simulate aerosol and radiation properties within the uncertainty ranges of the observations, defined in Table 2. Such a criterion is possible with synthetic observations because we know that the idealized truth is within the model uncertainty space, but the methodology would be more complex if we were using real observations. It is likely that some of the real observations will deviate from the model significantly because of model structural errors and issues related to the representativeness of the observations. Therefore, the use of real observations would necessitate the definition of a measure of plausibility that accounts for known structural and representativeness errors (McNeall et al., 2016; Williamson et al., 2013).

## 2.8 Aerosol effective radiative forcing (ERF)

We test the effect of observational constraint on the pre-industrial (PI, here 1850) to present-day (PD, 2008) July-mean European aerosol ERF and its components ERF$_{ACI}$ (Aerosol-Cloud Interaction) and ERF$_{ARI}$ (Aerosol-Radiation Interaction) as well as on the clear-sky component of the ERF$_{ARI}$ (ERF$_{ARIclr}$). The ERFs (except the ERF$_{ARIclr}$ term) account for above-cloud aerosol scattering and absorption (Ghan, 2013) and are calculated using a fixed sea-surface temperature from 2008.



## 3 Results

### 3.1 Relationships among the observed quantities and forcing variables

Figure 2 shows pairwise scatter plots of the PPE member output (Europe July-mean), which provides an overview of the spread of the model outputs as well as the relationships between the variables.

The aerosol variables show clear inter-relationships. In particular, AOD and $PM_{2.5}$ concentration show the strongest relationship (Pearson correlation, r = 0.88), which is expected given that satellite AOD measurements are frequently used as a proxy for ground-level $PM_{2.5}$ (Chu et al., 2016). This suggests that AOD and $PM_{2.5}$ observations will constrain the model uncertainty to a similar extent and therefore only one of these observable quantities is required. AOD and $PM_{2.5}$ are also clearly
correlated with sulphate, OC and BC, which are major components of $PM_{2.5}$ in polluted regions. CCN has a relatively weak positive relationship to both AOD (r = 0.46) and $PM_{2.5}$ (r = 0.21). A positive correlation is expected because, in general, greater aerosol loading will produce greater CCN concentrations, but the correlations are weak because the model aerosol size distribution (which determines CCN) can be configured in many different ways to produce the same AOD. The weak AOD-CCN relation has implications for model constraint: for example, AOD values in the range 0.15-0.2 encompass CCN
concentrations of around 400 to around 1000 $cm^{-3}$. These results are similar to those of (Stier, 2015) who showed similarly weak CCN-AOD correlations.

There are clear relationships between industrial-period forcing variables and some of the observable aerosol properties. For $ERF_{ARI}$ and $ERF_{ARIclr}$ the strongest relationships are with the sulphate concentration (r = -0.77 for $ERF_{ARIclr}$ and -0.67 for
$ERF_{ARI}$) and multi-decadal ΔAOD (r = 0.76 for $ERF_{ARIclr}$ and 0.55 for $ERF_{ARI}$). As expected, a present-day higher sulphate concentration corresponds to a stronger (more negative) $ERF_{ARI}$. ΔAOD is negative over Europe due to the reductions in anthropogenic aerosol emissions. Parameter settings that produce a strong multi-decadal ΔAOD also tend to produce a strong pre-industrial to present-day $ERF_{ARIclr}$ and therefore a stronger $ERF_{ARI}$. Based on these relationships, uncertainty in $ERF_{ARIclr}$ would be easier to constrain than uncertainty in $ERF_{ARI}$ and the most useful aerosol observation for this purpose would be
European-mean atmospheric sulfate concentration.

For the $ERF_{ACI}$ there is a relationship with the reflected shortwave ToA flux (r = -0.65), with a larger flux corresponding to a stronger (more negative) forcing. This relationship means that the parameter settings that produce more reflective aerosols and clouds in the present-day atmosphere also enhance $ERF_{ACI}$ forcing. There is also a relationship between the aerosol ERF
(preindustrial to present-day) and the 1978-2008 change in surface shortwave radiation (ΔSSR; r = -0.32). However, there is a lot of scatter in the relationship because the model parameters that cause uncertainty in decadal radiative changes are similar but not identical to those that cause uncertainty in forcing over the full industrial period (Regayre et al., 2018, 2014). The relationship between ΔSSR and aerosol ERF among eight models was used by Cherian et al. (2014) as an emergent constraint





on aerosol ERF over Europe. In section 4 we explore the implications of our uncertainty analysis for such emergent constraint studies.

In summary, the identified relationships in Figure 2 suggest that for Europe, constraints on sulphate concentration and ΔAOD could lead to some constraint on uncertainty in ERF$_{ARI}$ and ERF$_{ARIclr}$. Constraint on ToA flux could lead to some constraint on uncertainty in aerosol ERF and ERF$_{ACI}$ over Europe and observed multi-decadal changes in SSR could provide additional constraint. Observational constraints of ERFs are explored in section 3.5.

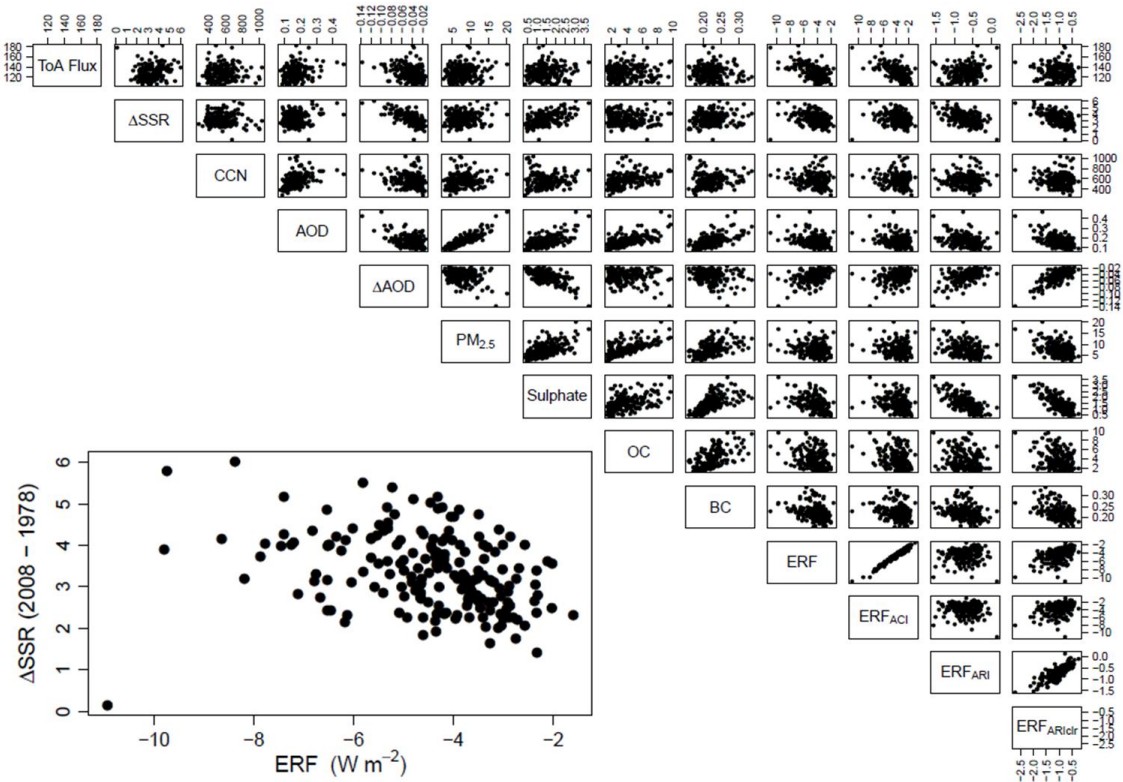

**Figure 2.** Pairwise scatter plots of the PPE member regional mean model output for Europe in July, for the aerosol properties used as constraints: ToA flux (W m$^{-2}$), change in SSR (ΔSSR, W m$^{-2}$) between 1978 and 2008, CCN conc. (cm-3), AOD, surface mass concentrations of **PM$_{2.5}$**, Sulphate, OC, and BC (μg m$^{-3}$), the changes in AOD (ΔAOD, W m$^{-2}$) between 1978 and 2008, and the 1850-2008 forcing variables: aerosol ERF, ERF$_{ACI}$, ERF$_{ARI}$ and ERF$_{ARIclr}$ (W m$^{-2}$).





## 3.2 Uncertainty in aerosols and radiative forcings

Figure 3 shows probability density functions of the observable aerosol quantities and the ERFs from the Monte Carlo sample of 4 million model variants (section 2.4). These pdfs sample the complete multi-dimensional parameter space of the model, weighted according the prior probability distributions on the input parameters (Yoshioka et al., 2018). The ranges are similar to those in Figure 2, but the PPE members themselves do not sample the parameter space densely enough to enable a statistically robust pdf to be generated.

The most uncertain observable aerosol properties, with the largest relative standard deviation (ratio of standard deviation to mean value) in our sample are the sulphate and OC concentrations and the multi-decadal $\Delta$AOD (Table 3). This suggests that constraining these properties will substantially reduce the sample size and constrain the parameter space. Some of the pdfs have long tails (e.g. OC concentration and $\Delta$AOD) which suggests that a subset of parameters may be combining in a specific manner to obtain these extreme values. The tails of the forcing pdfs contain the values most likely to be considered implausible against observations (Regayre et al., 2018).

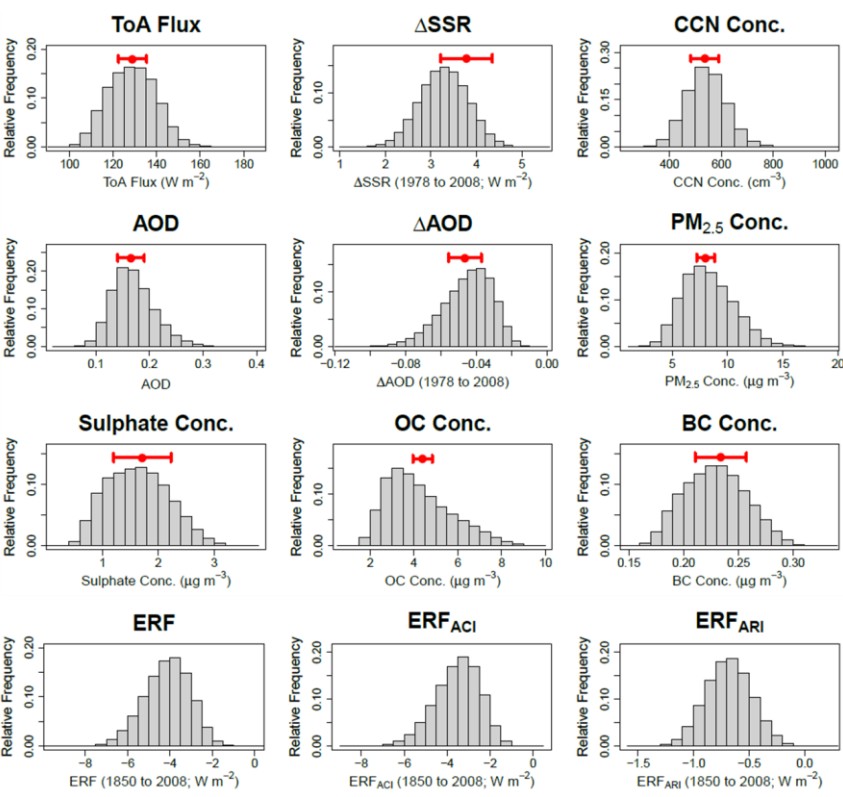



**Figure 3.** Calculated uncertainty in the aerosol quantities and aerosol ERF terms from the 4 million member sample. Results are for July-mean over Europe. The red bar shows the assumed range of each observation used to constrain the uncertain parameter space and the aerosol forcing uncertainty from Table 2.

| | mean | sd | \|sd/mean\| |
|---|---|---|---|
| Top of atmosphere upward SW flux (W m$^{-2}$) | 128.66 | 10.83 | 0.08 |
| Change in surface solar radiation from 1978 to 2008 (W m$^{-2}$) | 3.29 | 0.52 | 0.16 |
| Cloud condensation nucleus (CCN) conc. at 0.2% supersaturation (cm$^{-3}$) | 542.95 | 78.1 | 0.14 |
| Aerosol optical depth (AOD) | 0.17 | 0.04 | 0.23 |
| Change in AOD from 1978 to 2008, ΔAOD | -0.05 | 0.01 | 0.31 |
| **PM$_{2.5}$** mass conc. (µg m$^{-3}$) | 8.27 | 2.28 | 0.28 |
| Particle sulphate conc. (µg m$^{-3}$) | 1.63 | 0.54 | 0.33 |
| OC particle conc. (µg m$^{-3}$) | 4.25 | 1.48 | 0.35 |
| BC particle conc. (µg m$^{-3}$) | 0.23 | 0.03 | 0.12 |
| | | | |
| 1850 to 2008 ERF (W m$^{-2}$) | -4.14 | 1.07 | 0.26 |
| 1850 to 2008 ERF$_{ACI}$ (W m$^{-2}$) | -3.52 | 1.04 | 0.3 |
| 1850 to 2008 ERF$_{ARI}$ (W m$^{-2}$) | -0.68 | 0.2 | 0.3 |
| 1850 to 2008 ERF$_{ARIclr}$ (W m$^{-2}$) | -1.02 | 0.28 | 0.27 |

**Table 3:** Mean, standard deviation (sd) and absolute relative standard deviation (\|sd/mean\|) of the calculated uncertainty in

the aerosol quantities and aerosol ERF terms from the 4 million member sample. Results are for July-mean over Europe.

### 3.3 Sensitivity Analysis

We can decompose the overall variance in a model output into percentage contributions from the individual input parameters (Lee et al., 2011; Saltelli et al., 1999). The results of this analysis are shown in Figure 4.



For many of the output variables there is little correspondence with the forcing variables in terms of the main parameters that cause uncertainty. In particular, only about 10% of the CCN concentration uncertainty comes from the main causes of uncertainty in any of the corresponding forcing variables, which is consistent with the weak correlations in Figure 2 and the conclusions of Lee et al. (2016).

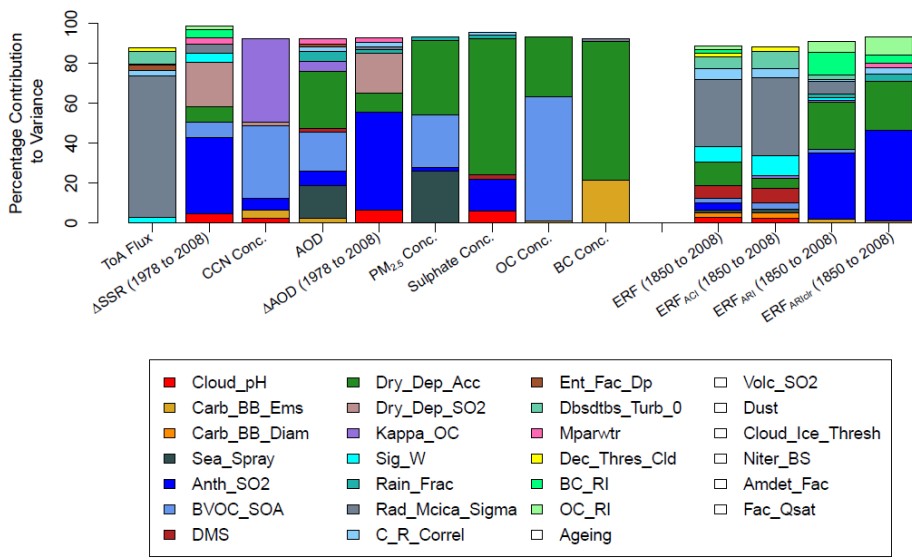

**Figure 4.** Variance-based sensitivity analysis results, showing the percentage parameter contributions to model output uncertainty in the observable aerosol quantities and the forcing variables for Europe. Only those parameters which cause at least 1% of the variance are shown in colour.

There is reasonable correspondence between the sources of uncertainty in sulfate concentration, $\Delta$AOD, ERF$_{ARI}$ and ERF$_{ARIclr}$, which is again consistent with Figure 2. Around 60-70% of the output variance in these variables is accounted for by anthropogenic SO$_2$ emissions (Anth_SO2) and the accumulation mode dry deposition velocity (Dry_Dep_Acc). This degree of similarity in the parametric uncertainty sources implies that an individual observational constraint on $\Delta$AOD should lead to

15 some constraint of the ERF$_{ARI}$ and ERF$_{ARIclr}$ forcing uncertainty. Dry_Dep_Acc is also a significant cause of uncertainty in AOD, PM$_{2.5}$ and concentrations of sulphate, OC and BC (between ~25% and ~70% for each). Hence, it is possible that constraint of these observable aerosol quantities may lead to some constraint on ERF$_{ARI}$ and ERF$_{ARIclr}$ uncertainty. We also see that the uncertainty in the ToA flux is dominated by the cloud radiation parameter Rad_Mcica_Sigma parameter (which affects the spatial homogeneity of the clouds), which also accounts for about 35% of the variance in ERF and ERF$_{ACI}$. This

parameter also causes most of the uncertainty in global mean ERF and ToA flux (Regayre et al., 2018). However, over Europe there are multiple other parameters causing a small amount of the aerosol ERF uncertainty which suggests an effective constraint will require using multiple complementary observations.



### 3.4 Constraint of aerosol properties

We first explore how constraining an individual aerosol property helps to constrain the range of other observable properties and multi-decadal trends. AOD is the aerosol property most frequently observed and used to evaluate and constrain models (e.g., Shindell et al. (2013)) and is used as the control variable in data assimilation used to evaluate the aerosol forcing (Bellouin

et al., 2013). Figure 5 shows the reduction in uncertainty of the modelled atmospheric properties and trends by constraining the model to match observed AOD. Credible intervals (95%) corresponding to the individual constraints are provided in Table 4.

Constraint of European monthly-mean AOD to lie in the range 0.14-0.19 (23% of the full ensemble range) leads to a fairly strong constraint of $PM_{2.5}$ uncertainty: the standard deviation of $PM_{2.5}$ ($\sigma PM_{2.5}$) is reduced by 34% when the range of AOD is

reduced by about 77%. The standard deviation of the PM components and the multi-decadal trend $\Delta AOD$ are also reduced, but by a smaller amount: around 20% for OC and only around 10% for BC, sulphate and $\Delta AOD$. Constraint of individual chemical components is weaker because there are many combinations of sulphate, BC and OC that can account for high or low AOD. Uncertainty in the other observable quantities (CCN and ToA flux and the multi-decadal trend $\Delta SSR$) are essentially unaffected by the constraint of AOD. The reason for the weak constraint is that there are many model variants within the

observed range of AOD (or $PM_{2.5}$) that produce very different CCN, ToA flux and $\Delta SSR$.

These results provide some indication of the possible remaining uncertainty in a model that has been tuned to agree with AOD observations. A tuned model that agrees with AOD observations within the observational uncertainty is just one of many potential variants of the model that have equally good agreement with the observations. For example, our model suggests that the remaining uncertainty (absolute range) in European-mean CCN could be 755 cm$^{-3}$ in a model constrained by AOD

observations, which is only slightly less than the unconstrained range of 782 cm$^{-3}$. Most surprisingly, constraint of AOD leaves open a wide range of potential values of the change in AOD over decadal periods. The range of $\Delta AOD$ from 1978-2008 after constraint of 2008 AOD is 0.105 (range -0.109 to -0.004), which is only slightly lower than the unconstrained range of 0.115. Screening model variants based on their ability to reproduce a single aerosol-related observation is not a sufficient constraint on aerosol-related model uncertainty. Therefore tuning a model to AOD observations is completely inadequate for producing

a robust aerosol model.





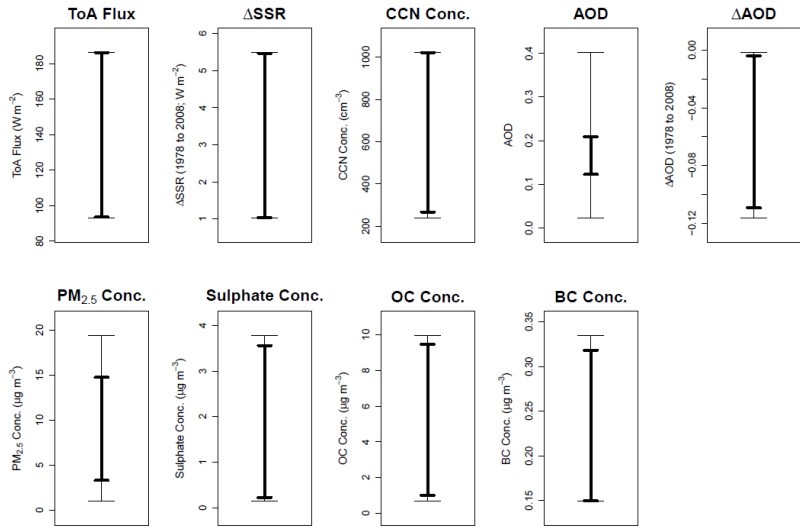

**Figure 5.** Effect of observational constraint of AOD on other aerosol properties in the model over Europe. The bars show the absolute range of the pdf before (thin line) and after (thick line) constraint. Results are for mean properties over Europe in July.

| | 95th CI unconstrained | 95th CI constrained by AOD |
|---|---|---|
| Top of atmosphere upward SW flux (W m$^{-2}$) | (108.9 , 149.5) | (109.0 , 149.0) |
| Change in surface solar radiation from 1978-2008 (W m$^{-2}$) | (2.27 , 4.30) | (2.29 , 4.30) |
| Cloud condensation nucleus (CCN) conc. at 0.2% supersaturation (cm$^{-3}$) | (396 , 704) | (408 , 698) |
| Aerosol optical depth (AOD) | (0.105 , 0.257) | (0.130 , 0.201) |
| Change in AOD from 2008 – 1978, ΔAOD | (-0.076 , -0.022) | (-0.072 , -0.024) |
| **PM$_{2.5}$** mass conc. (μg m$^{-3}$) | (4.41 , 13.15) | (5.31 , 10.98) |
| Particle sulphate conc. (μg m$^{-3}$) | (0.71 , 2.72) | (0.79 , 2.58) |
| OC particle conc. (μg m$^{-3}$) | (2.10 , 7.61) | (2.24 , 6.66) |
| BC particle conc. (μg m$^{-3}$) | (0.179 , 0.284) | (0.183 , 0.277) |

**Table 4.** Effect on the uncertainty in aerosol properties over Europe when the model is constrained by **PM$_{2.5}$** measurements (assumed measurement uncertainty range 7.2-8.8 μg m$^{-3}$) or AOD (assumed measurement uncertainty range 0.14-0.19). The aerosol uncertainties are given as the 2.5th and 97.5th empirical percentiles of the pdf to form a 95% credible interval.





### 3.5 Constraint of 1850 – 2008 aerosol ERF uncertainty

#### 3.5.1 Effects of individual aerosol and radiation observational constraints

Figure 6 (top row) shows the potential constraint achievable on uncertainty in the 1850–2008 aerosol ERF, $ERF_{ACI}$, $ERF_{ARI}$ and $ERF_{ARIclr}$ when we constrain July-mean AOD over Europe. Each box and whisker plot shows the uncertainty distribution

from the original sample of 4 million model variants (grey, left) and the sample of constrained models (pink, right). Table 5 shows means and standard deviations for the original and constrained distributions from AOD and all other individual observational constraints.

Observational constraint of simulated AOD has essentially no effect on the range of aerosol ERF and the $ERF_{ACI}$ component

of forcing over Europe. There is some reduction in uncertainty in the $ERF_{ARIclr}$ component of forcing (standard deviation reduced by around 12%) but not in $ERF_{ARI}$, despite both sharing common causes of uncertainty with AOD (Section 3.3).

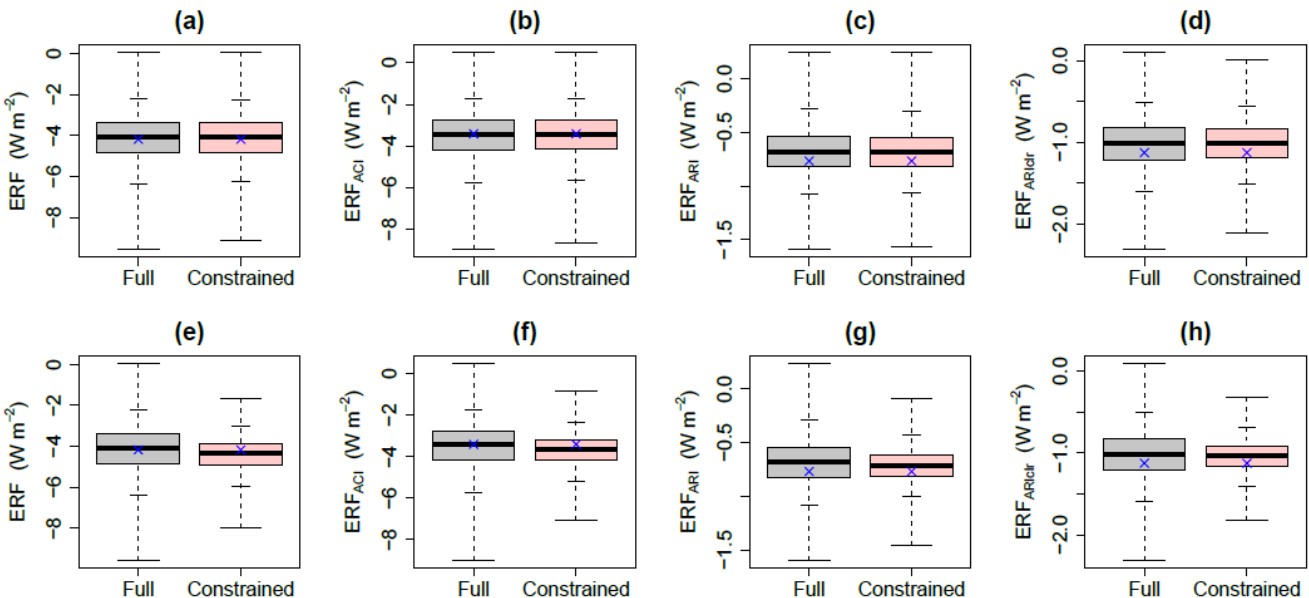

**Figure 6.** The effect of the AOD constraint (top row) and all observational constraints together (bottom row) on the uncertainty
in the 1850-2008 forcing variables over Europe. Columns left to right show constraint of aerosol ERF, $ERF_{ACI}$, $ERF_{ARI}$ and $ERF_{ARIclr}$ (W m$^{-2}$) respectively. The boxes show the inter-quartile range (with the median value shown by the black line that cuts it) and the whiskers show the full range of the distribution. The short horizontal bars on the whiskers correspond to 95% credible interval bounds. The grey boxes show the distribution for the variable predicted over the original sample (4 million model variants that span the underlying parameter uncertainty) and the pink boxes show the corresponding distribution of the
remaining samples after the constraint has been applied. The predicted forcing using the input combination of the model run used as the idealized observation is shown by the blue cross.



| Applied constraint | ERF (W m⁻²) | ERF$_{ACI}$ (W m⁻²) | ERF$_{ARI}$ (W m⁻²) | ERF$_{ARIclr}$ (W m⁻²) |
|---|---|---|---|---|
| **No constraint** | -4.144  (1.075) | -3.523  (1.044) | -0.683  (0.204) | -1.024  (0.281) |
| | | | | |
| **All constraints** | -4.391  (0.759) | -3.707  (0.736) | -0.710  (0.148) | -1.044  (0.184) |
| | | | | |
| **ToA Flux** | -4.137  (0.863) | -3.496  (0.790) | -0.683  (0.201) | -1.024  (0.280) |
| **ΔSSR (1978-2008)** | -4.247  (1.045) | -3.593  (1.034) | -0.710  (0.193) | -1.067  (0.260) |
| **CCN Conc.** | -4.181  (1.065) | -3.547  (1.036) | -0.695  (0.204) | -1.039  (0.277) |
| **AOD** | -4.123  (1.034) | -3.500  (1.017) | -0.684  (0.194) | -1.014  (0.246) |
| **ΔAOD (1978-2008)** | -4.175  (1.003) | -3.541  (1.011) | -0.693  (0.176) | -1.034  (0.214) |
| **PM$_{2.5}$ Conc.** | -4.173  (1.057) | -3.541  (1.039) | -0.688  (0.197) | -1.021  (0.257) |
| **Sulphate Conc.** | -4.231  (1.037) | -3.589  (1.040) | -0.695  (0.167) | -1.035  (0.222) |
| **OC Conc.** | -4.245  (1.047) | -3.622  (1.027) | -0.682  (0.207) | -1.018  (0.278) |
| **BC Conc.** | -4.263  (1.058) | -3.607  (1.046) | -0.707  (0.195) | -1.052  (0.259) |

**Table 5:** Mean and standard deviation (in brackets) of the forcing distributions over Europe for the original unconstrained sample, the multiple-constraint sample and the individually constrained samples where each observational constraint is applied independently.

Figure 7 summarises the effect of the other individual constraints. For ERF$_{ACI}$ (and therefore aerosol ERF, which is dominated by ACI) the only observation that has any meaningful effect on the range is the ToA flux. When the flux is constrained to be within the range 122-135 W m⁻² (from the prior range of 90-175 W m⁻²) the standard deviation of ERF$_{ACI}$ over Europe falls by 24% (Table 5). The ΔSSR observation reduces the aerosol ERF and ERF$_{ACI}$ standard deviations by less than 3%. The only other constraints on uncertainty in aerosol ERF and ERF$_{ACI}$ come from constraining AOD or ΔAOD, which reduce the forcing uncertainties by around 3% each. The effect of applying all observations in combination is discussed in Section 3.5.2.

ERF$_{ARI}$ and ERF$_{ARIclr}$ are constrained by several individually applied observations. ΔAOD and sulphate concentrations provide the strongest constraints. ΔAOD reduces the standard deviation of ERF$_{ARI}$ and ERF$_{ARIclr}$ by 14% and 24% respectively. Constraining sulphate concentrations reduces the uncertainty in ERF$_{ARI}$ by 18% and in ERF$_{ARIclr}$ by 21%. The strong constraint of ERF$_{ARI}$ and ERF$_{ARIclr}$ uncertainty is consistent with Figure 4, where we saw that around 60-70% of the uncertainty in each of ΔAOD, ERF$_{ARI}$ and ERF$_{ARIclr}$ could be attributed to the same two parameters. Again, the relatively weak constraint is caused by interacting combinations of parameter effects ((Lee et al., 2016; Regayre et al., 2018); Section 3.7), so there is potential for





significant error compensation (or equifinality, Beven and Freer, 2001). In all other cases the individual observational constraints reduce the uncertainty in ERF$_{ARI}$ and ERF$_{ARIclr}$ by less than 10%.

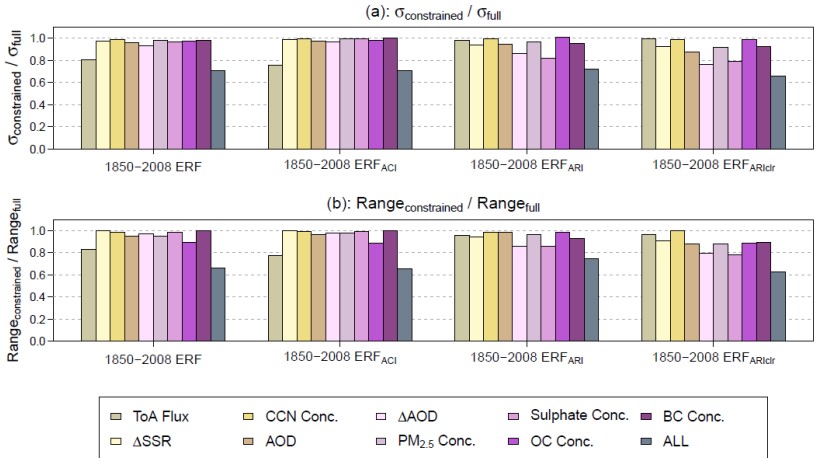

**Figure 7.** The relative constraint achieved for aerosol ERF, ERF$_{ACI}$, ERF$_{ARI}$ and ERF$_{ARIclr}$ over Europe given the individual constraints applied (colours) and the simultaneous constraint (ALL). The relative constraint is evaluated as the ratio of the standard deviation of the forcing in the constrained sample ($\sigma_{constrained}$) to the standard deviation of the forcing in the original, unconstrained sample ($\sigma_{full}$).

### 3.5.2 Effect of all observational constraints

Figure 6 (bottom row) and the right-most bars in Figure 7 show the reduction in Europe-mean 1850 – 2008 aerosol ERF, ERF$_{ACI}$, ERF$_{ARI}$ and ERF$_{ARIclr}$ uncertainty when we simultaneously apply all nine observational constraints. The standard deviations are reduced by 29.4% for the aerosol ERF, 29.5% for ERF$_{ACI}$, 27.8% for ERF$_{ARI}$ and 34.3% for ERF$_{ARIclr}$ (Table 5) and Figure 6 shows a reduction in the interquartile range (box width) and 95% credible interval in each case.

Our results show that multiple observational constraints are very effective at reducing the plausible parameter space (ruling out 96.4% of model variants). However, these reductions in parameter space have only a modest impact on aerosol ERF uncertainty. This occurs because the 27 parameter values in the constrained space can be combined to produce a wide range of ERFs (Lee et al. 2016). These results highlight the value of exploring the wider underlying modelling uncertainties (achieved here using a well-designed PPE to inform statistical emulation). The more comprehensive exploration of the parameter space

using several million model variants from the emulators enabled us to explore the wider uncertainties that would not have been captured even by the 191 PPE members. Furthermore, a 96.4% reduction in parameter space would have reduced the number of PPE members to one or two, which would not have revealed that a large fraction of the ERF uncertainty (70.6%) remained unconstrained. Likewise, a single model variant arrived at through tuning cannot represent model behaviour over the remaining



plausible parameter space. Similar concerns about non-robust samples apply also to the small number of members in multi-model ensembles.

### 3.5.3 Effect of combinations of observational constraints

An important question in model constraint is how quickly the model uncertainty falls as additional observational constraints are applied. Figure 8 shows the average reduction in forcing uncertainty versus the number of observational constraints applied. With nine possible observational constraints there are nine possible single constraints, 36 possible combinations of 2 constraints, 252 combinations of 3 constraints, etc. We therefore show a mean over all possible combinations of each number of constraints.

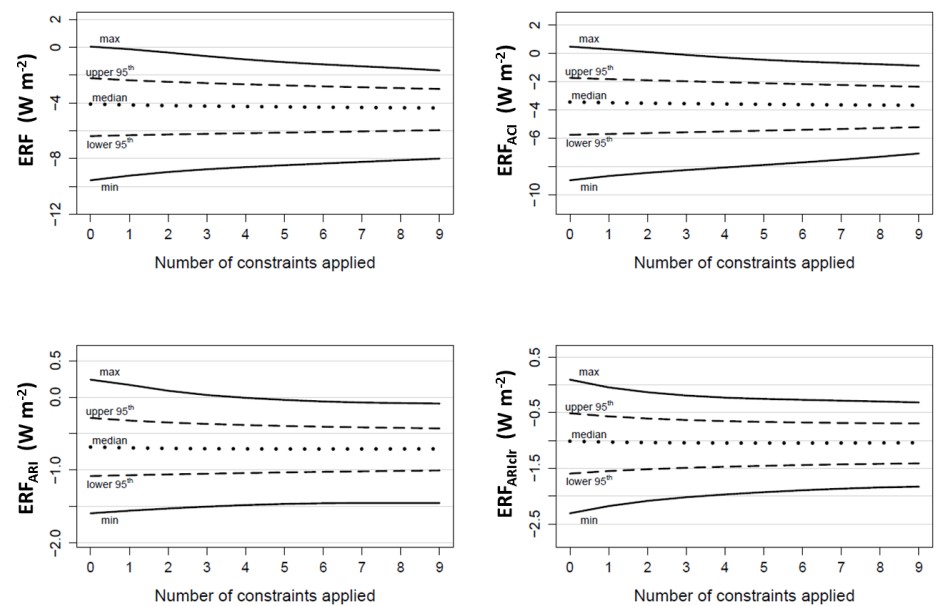

**Figure 8.** The dependence of aerosol ERF uncertainty on the number of observational constraints applied. The lines show the mean effect of different numbers of constraints.

Averaged across the many combinations of constraints, uncertainty in aerosol ERF and its components initially falls approximately linearly with the number of constraints applied, but then flattens out. This dependence implies that some observations are constraining the same sources of uncertainty as other observations (as shown in section 3.3). So while a large number of observations are needed to constrain forcing, it is also important to identify observations that provide unique constraints on parameter uncertainties. The effectiveness of each observational constraint depends on which other constraints are applied with it. For example, two positively correlated observations like $PM_{2.5}$ and AOD (Figure 2) will reduce the





allowable parameter space in broadly the same dimensions because the same parameters cause their uncertainty (Figure 4). Therefore the constraint on forcing uncertainty achieved by AOD and PM$_{2.5}$ is not additive.

### 3.6 Constraint of 1978 – 2008 forcing uncertainty

Previous research has shown that the causes of uncertainty in recent decadal forcing are quite different to the causes of uncertainty in pre-industrial to present-day forcing (Regayre et al., 2018, 2014). Much of the uncertainty in PI to PD aerosol–cloud interaction forcing is caused by natural aerosols (Carslaw et al., 2013; Carslaw et al., 2017), which are much less important over recent decades. We therefore expect recent aerosol and radiation observations to provide a greater constraint on recent decadal forcings than on forcing referenced to PI conditions. Our results show that this hypothesis is correct: simultaneous application of the nine observational constraints reduces the standard deviation of the 1978-2008 aerosol forcing distributions by 33.7% for ERF, 32.3% for ERF$_{ACI}$, 35.0% for ERF$_{ARI}$ and by 43.9% for ERF$_{ARIclr}$, which are all greater reductions than for the 1850 to 2008 forcing (Figure 7, Table 5). The main contributor to the reduction in uncertainty in the aerosol ERF from 1978 to 2008 is the change in AOD, followed by present-day AOD. These results suggest that forcing uncertainty in recent decades may be more readily constrained by observations than multi-century forcing.

### 3.7 Constraint of plausible parameter ranges

The overall objective of our approach is to identify all the observationally plausible variants of the model so that they can be used to calculate an observationally constrained spread of aerosol ERFs. Each variant is associated with a particular part of parameter space. We can then use the emulators to compute the constrained magnitude and range of any other aerosol property (or the changes between 1850, 1978 and 2008). Alternatively a sample of these variants (parameter settings) could be used in the model itself to simulate aerosol effects for any situation (for example, with very different meteorological conditions, or anthropogenic aerosol emissions).

Figure 9 shows a one-dimensional projection of the remaining parameter space after constraining to the nine observations. There are some substantial reductions in the plausible marginal range of several individual parameters. It needs to be borne in mind that, with 27 parameter dimensions, the parameter relationships which have been constrained by multiple observations cannot be seen in the one-dimensional projection. That is, the remaining plausible individual parameter values can be combined in many ways with the remaining space of the other parameters and still reproduce all of the observations (Lee et al., 2016; Regayre et al., 2018). Figure 9 identifies parts of parameter space that are absolutely ruled out. For example, a very low setting of the BVOC_SOA parameter cannot produce observationally plausible results when combined with any of the possible combinations of the other 26 parameters.





The constraint of the parameter ranges will be different when using real observations, but it is interesting to see how nine observations can marginally constrain twenty-seven parameters when there is a high degree of potential compensating effects. The strongest marginal constraint is on: the sea spray aerosol emissions (Sea_Spray; the highest 25% and lowest 15% of values are implausible), biogenic secondary aerosol formation (BVOC_SOA; the lowest 40% and top 20% of the range are implausible), the hygroscopicity of organic carbon (Kappa_OC; the top 40% of values are implausible), and the imaginary part of organic carbon refractive index (OC_RI; top 30% is implausible). Furthermore, the lowest 10-20% of the range of several aerosol emission parameters are also deemed implausible (biomass burning (Carb_BB_Ems), degassing volcanic (Volc_SO2), DMS, anthropogenic sulphur dioxide (Anth_SO2)).

The atmospheric (host model) marginal parameter ranges are much less constrained because the observable variables that we used are not strongly dependent on them, except for ToA flux observations which are known to be affected by the Dec_Thresh_Cld and Rad_Mcica_Sigma parameters (Regayre et al., 2018). Values of the threshold for cloudy boundary layer decoupling parameter (Dec_Thresh_Cld) are concentrated towards the lower end of the range (the upper 40% are implausible). We also show that the top 20% of values are implausible for the parameter controlling the amount of overlap between sub-grid clouds as seen by the model's radiation code (Rad_Mcica_Sigma). However, the lowest 40% of this parameter range can be entirely ruled out by constraining the ToA flux in the North Pacific (Regayre et al., 2018). These results highlight the important benefits which will come from constraining the model uncertainty using multiple observations in multiple environments.

This analysis highlights the complexity of the multi-dimensional parameter uncertainty space that remains after observational constraint: there are clearly a large number of ways of tuning a model to be observationally plausible.





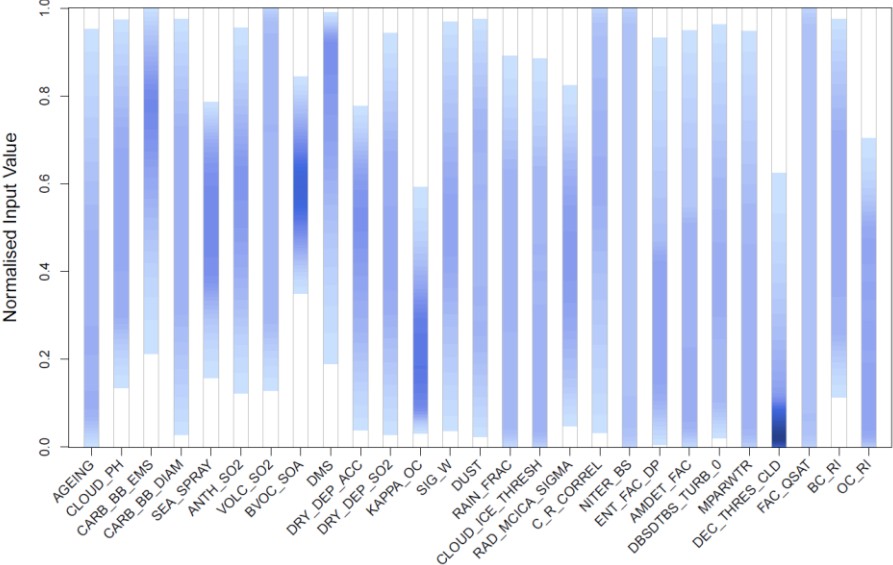

**Figure 9.** One-dimensional projection of the remaining parameter space after simultaneous constraint of all atmospheric quantities and decadal trends.

## 4   Implications for model screening and emergent constraints

In multi-model ensembles it is usually the case that each modelling group submits a single well-tuned version (variant) of a model. The uncertainty in the ensemble is determined by the structural differences between the models, but the uncertainty in the individual models (caused by multiple uncertain parameter settings) is not quantified. Here we use the uncertainty in HadGEM3-UKCA to estimate the effect it might have on the results of multi-model emergent constraint studies. Clearly the uncertainties in each model will differ, so we use our model uncertainty only as a rough estimate of the potential effect.

In the ACCMIP study (Shindell et al., 2013) model skill at simulating AOD was used to screen nine models. We have described in Sections 3.4 and 3.5 why constraint of AOD can only be considered the first step in model screening; AOD does not effectively reduce model uncertainty when used in isolation. The standard deviation of the modelled global annual mean $ERF_{ARI}$ in the ACCMIP study was about 50% of the multi-model mean. In our results, after we have screened out model variants that are inconsistent with synthetic AOD observations (i.e., effectively tuning to AOD), the standard deviation of the HadGEM-UKCA $ERF_{ARI}$ over Europe is about 30% of our mean. Therefore the standard deviation in HadGEM3-UKCA caused by uncertain input parameters is a significant fraction of the multi-model standard deviation, and would affect the constrained range of $ERF_{ARI}$. Shindell et al. (2013) acknowledged that uncertainties in the emissions could alter the relative agreement of the models with observations and thereby affect the spread of plausible model predictions. However, uncertainty in emissions is just one of many possible sources of uncertainty that could affect the conclusions (Figure 4).



In emergent constraint studies a linear relationship between aerosol forcing and an observable variable simulated by multiple models is used to define an observationally constrained value of the variable of interest. In the Cherian et al. (2014) study Europe-mean aerosol ERF was estimated by regressing modelled ERF against the 1990-2005 modelled trend in SSR over Europe from seven aerosol-climate models. An observed SSR trend of -4.0 ± 0.6 W m$^{-2}$ decade$^{-1}$ enabled the Europe-mean

aerosol ERF to be constrained to -3.56 ± 1.41 W m$^{-2}$ (corresponding to the range of -4.97 to -2.15 W m$^{-2}$). Their analysis accounted for the uncertainty in SSR caused by meteorological variability but did not account for the influence of parametric uncertainties.

In our HadGEM3-UKCA simulations the Europe-mean aerosol ERF 95% credible interval is -6.0 to -2.7 W m$^{-2}$ after constraining the model using nine observations (i.e., a tight tuning of the model). This range provides some measure of the

range of alternative ERFs that could be obtained by the individual models had they been tuned differently (*but equally well*) to observations (although we do not know what actual tuning was undertaken). Our single-model uncertainty range is comparable to the multi-model ensemble range, but was not accounted for by Cherian et al. (2014) in deriving the emergent constraint. The effect of including this previously neglected source of single-model uncertainty is to substantially increase the uncertainty on the emergent constraint (Figure 10). Furthermore, the likely magnitude of forcing derived from emergent

constraints is sensitive to the uncertainties accounted for in the process (Samset et al., 2014).

In many emergent constraint studies, the constrained ERF (or other quantity) is essentially based on the very small number of models that lie within the uncertainty range of one observation (Figure 10). With our approach, model variants that are plausible against this one observation type are then examined to determine their plausibility against many other observation types – in this study, nine observations in total. Ultimately, multivariate constraint is essential to reach robust conclusions because of the

many compensating sources of model uncertainty.

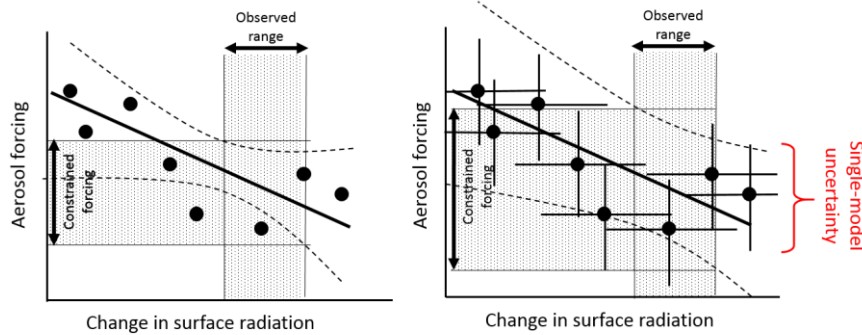

**Figure 10.** An example of an emergent constraint of the aerosol forcing using results from multiple models. (a) a relatively
tight apparent constraint when the uncertainty in the individual models is neglected; (b) a much weaker constraint when the uncertainty in the individual models is accounted for.





# 5    Conclusions

The use of observations to produce a well-configured model variant is a fundamental aspect of ensuring that models can make trustworthy predictions. For example, in a review of progress on reducing uncertainty in direct radiative forcing Kahn (2012) argues that models can be "constrained by the aggregate of observational data, to calculate the regional and global radiation fields and material fluxes with adequate space–time resolution to produce the best result we can achieve." The primary objective of our study was to determine how much uncertainty could remain in an aerosol-climate model when it is constrained to match combinations of observations that define the base state of the model: top-of-atmosphere upward shortwave flux, aerosol optical depth and its decadal change, $PM_{2.5}$, cloud condensation nuclei, concentrations of sulphate, black carbon and organic material as well as multi-decadal change in surface shortwave radiation and aerosol optical depth. Our results refer to July-mean conditions over Europe.

To estimate the uncertainty that might typically exist in a climate model before and after tuning, we used a perturbed parameter ensemble that sampled uncertainties in 27 parameters related to aerosol emissions, aerosol and cloud processes, and parameters in the host physical climate model that influence clouds, humidity, convection and radiation in the base state of the model. We performed 191 model simulations that spanned the 27-dimensional space of the model uncertainty and then built surrogate model emulators from which we created a Monte Carlo sample of 4 million 'model variants'. Constraining the model outputs using all nine observations rules out over 96% of the model variants and the associated implausible parts of parameter space. The remaining 153,000 model variants have been used to estimate the observationally constrained aerosol ERF and the uncertainty associated with one tuned version of HadGEM3-UKCA.

Tuning HadGEM3-UKCA to AOD alone has almost no effect on the reliability of the tuned model to simulate CCN (and hence cloud drop number concentrations) (Figure 5). Constraining European-mean AOD in July to lie within a realistic range of 0.14 – 0.19 (23% of the full model uncertainty range) results in a reduction of less than 5% in the uncertainty in CCN generated by the full set of 4 million variants of the model. The CCN range is then 268-1022 $cm^{-3}$ compared to 241 – 1022 $cm^{-3}$. This provides a measure of the parametric uncertainty when AOD measurements are used to infer CCN, although the range would potentially be larger had we perturbed more parameters (Yoshioka et al., 2018). Tuning a model to AOD alone also has very little effect on the modelled range of the trend in AOD over a multi-decadal period. The complete set of model variants produces a range of 1978 to 2008 changes in AOD over Europe of 0.115, and this range is only reduced to 0.105 (range –0.109 to 0.004) when the model is constrained to match European-mean 2008 AOD within observational uncertainty.





Constraint of AOD alone has a negligible effect on the uncertainty in the aerosol ERF over Europe. Although the aerosol ERF simulated by a model will change as parameters are tuned to achieve agreement with AOD measurements, any resulting ERF will have large uncertainty (i.e., there are many other equally well-tuned model variants that produce different ERFs). This uncertainty cannot easily be estimated without a full uncertainty analysis of the model as we have done here. The weak

constraint calls into question the robustness of estimates of aerosol forcing based on AOD reanalyses (Bellouin et al., 2013)

Observational constraint using nine observations has the potential to reduce the uncertainty in aerosol ERF: the standard deviation falls by 29.4% for the 1850-2008 aerosol ERF, 29.5% for $ERF_{ACI}$, 27.8% for $ERF_{ARI}$ and 34.3% for $ERF_{ARIclr}$. The standard deviation of 1978-2008 aerosol ERF could be reduced by around 34%, which is greater than for the 1850-2008 ERF because there is greater correspondence between the causes of uncertainty in near-term aerosol forcing and the 2008 aerosol-

cloud-radiation state than there is between the 1850-2008 ERF and the 2008 state (Regayre et al., 2018, 2014). Because near-term future changes in aerosols and clouds are likely to resemble recent changes more than centennial-scale changes, we are optimistic that the uncertainty in near-term aerosol ERFs could be constrained and used to provide policy-relevant information on near-term temperature changes (Hawkins et al., 2017).

The most effective observational constraint on the uncertainty in aerosol ERF and $ERF_{ACI}$ is the top-of-atmosphere upward

shortwave flux. When the flux is constrained to be within the range 122-135 W m$^{-2}$ (from the prior range of 90-175 W m$^{-2}$) the standard deviation of $ERF_{ACI}$ over Europe falls by 24%. Other observational constraints reduce the $ERF_{ACI}$ uncertainty by a few percent at most, including the change in surface SW radiation. Effectively, this result means that routine tuning of radiative fluxes in climate models will have a bearing on the magnitude of the aerosol ERF that the models calculate. The reason for the constraint on forcing uncertainty is that model parameters that control cloud and atmosphere brightness also control how that

brightness responds to changes in aerosols over Europe. However, it is only likely to be an effective constraint where the brightness is controlled by tuning the parameters we have identified here. In regions dominated by quite different processes, like mixed-phase clouds, tuning the flux will have a much weaker effect on the aerosol ERF.

The most effective observational constraints on the uncertainty in $ERF_{ARI}$ and $ERF_{ARIclr}$ over Europe are the sulphate

concentration and the change in AOD over a multi-decadal period (we used 1978-2008). When applied individually, sulphate concentrations constrain $ERF_{ARI}$ standard deviation in our ensemble by 18% over Europe. The 1978-2008 change in AOD constrains the $ERF_{ARI}$ standard deviation by 14% when applied individually. Constraint of AOD itself (in 2008) reduces the $ERF_{ARI}$ uncertainty by only 5%, and would not provide a realistic way of screening models. The other constraints were much less effective.

The plausible ranges of some natural aerosol emissions are reduced after constraining to the nine observations, particularly sea spray emissions and biogenic volatile organic aerosol formation. We were also able to constrain some aerosol process parameters such as the CCN hygroscopicity (kappa), the imaginary refractive indices of BC and OC, and parameters controlling





boundary layer stability and the radiative properties of overlapping sub-grid clouds which control cloud brightness. Observational constraint generates a set of constrained parameter settings that can be taken forward and used to make model predictions under any other conditions (e.g. for future projections).

The range and combinations of observationally plausible parameter values remain very large even after constraint using nine observations, which explains why the aerosol ERF uncertainty remains large after constraint. This result is not a failure of our approach, but rather an indication of the multiple ways in which uncertain model parameters can combine to predict a wide range of outputs with equal skill when compared to observations. These multiple model variants are neglected when a single model variant is produced through tuning.

Widely used procedures of aerosol-climate model evaluation and observational 'validation' lack statistical robustness because they do not adequately sample the model uncertainty space. We showed that observational constraint against nine observations identified less than 4% of the 4 million sampled points in multi-dimensional parameter space as plausible (i.e., the model value is within the observational uncertainty). A 96% reduction in parameter space would have reduced our original set of 191

ensemble members to one or two, which would not have revealed that a large fraction of the ERF uncertainty (about 71%) remained unconstrained. This creates a fundamental problem for multi-model ensembles (which have far fewer members) and model tuning (which may explore only a few dozen model variants and mostly with single parameter perturbations). From such small samples of models it is not possible to determine how observations help to reduce model uncertainty, so estimates of radiative forcing should not be considered robust.

Our results have implications for studies that seek emergent constraints on a small set of models based on one observational metric. An emergent constraint can be informative, but cannot be expected to reduce the uncertainty in a complex model when used in isolation. The example closest to our study is Cherian et al. (2014) in which the relationship between aerosol ERF and the trend in surface solar radiation (SSR) over Europe for seven climate models was used to estimate the observationally

constrained uncertainty in aerosol ERF. Our results for the HadGEM3-UKCA model show that the uncertainty in aerosol ERF and SSR trend in any one tuned version of the model is likely to be of the same order of magnitude as the multi-model range. If the uncertainties in individual models in an ensemble are not accounted for, then we risk being over-confident in the emergent constraints. Efforts to quantify and observationally constrain individual models are therefore not an alternative to multi-model studies, but individual model uncertainty needs to be quantified and incorporated as an essential component of such efforts to

understand and then reduce aerosol ERF uncertainty.

There is considerable scope to extend our approach to incorporate more observation types and more regions. These should include: 1) Aerosol and radiation trends (Allen et al., 2013; Cherian et al., 2014; Leibensperger et al., 2012; Li et al., 2013; Liepert and Tegen, 2002; Shindell et al., 2013; Turnock et al., 2015; Zhang et al., 2017). So far we used changes in AOD and



SSR, but changes in ToA SW flux as well as aerosol components like OC (Ridley et al., 2018) and sulfate could provide useful constraints. 2) Observations from pristine regions that might provide a constraint on preindustrial-like aerosol and cloud properties (Carslaw et al., 2017; Hamilton et al., 2014). 3) Information on the vertical profile of aerosols. 4) Observed relationships between changes in aerosol and cloud variables (Ghan et al., 2016; Gryspeerdt et al., 2017b; Quaas et al., 2009)

such as defined in Eq. 1. Such relationships are a favoured way to constrain forcing. Although it is conceivable that relationships between change-of-state variables can be predicted more reliably than state variables themselves (because of cancellation of correlated model errors), the model uncertainty in these relationships has not been determined in studies that have applied them.

Whichever approach is used to reduce uncertainty in aerosol forcing, it is essential to acknowledge that aerosol-chemistry-climate models are highly complex with dozens of sources of uncertainty that can be combined in many ways. Such a system cannot be constrained by one or two observations at a time, and emergent constraints are no different in this respect. Robust constraint of a high-dimensional system requires large numbers of combined constraints so that the multiple compensating dimensions of uncertainty can be reduced (Reddington et al., 2017). We are reasonably confident that extension of our approach

to more and varied observations will enable the uncertainty in aerosol radiative forcing to be reduced significantly.

**Acknowledgements**

This research was funded by the Natural Environment Research Council (NERC) under Grants NE/J024252/1 (GASSP), NE/I020059/1 (ACID-PRUF) and NE/P013406/1 (A-CURE); the European Union ACTRIS-2 project under grant 262254; the National Centre for Atmospheric Science (Yoshioka, Carslaw); and by the UK–China Research and Innovation Partnership

Fund through the Met Office Climate Science for Service Partnership (CSSP) China as part of the Newton Fund. We made use of the N8 HPC facility funded from the N8 consortium and an Engineering and Physical Sciences Research Council Grant to use ARCHER (EP/K000225/1) and the JASMIN facility (www.jasmin.ac.uk/) via the Centre for Environmental Data Analysis funded by NERC and the UK Space Agency and delivered by the Science and Technology Facilities Council. We acknowledge the following additional funding: the Royal Society Wolfson Merit Award (Carslaw); a doctoral training grant

from the Natural Environment Research Council and a CASE studentship with the Met Office Hadley Centre (Regayre).

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
