# Peer review of "The importance of comprehensive parameter sampling and multiple observations for robust constraint of aerosol radiative forcing"

_Atmospheric Chemistry and Physics, 2018_

## Referee Comment (RC1) · J. Mülmenstädt (Referee) · 18 Apr 2018

Dear Patrick,

I have reviewed "The importance of comprehensive parameter sampling and multiple observations for robust constraint of aerosol radiative forcing" by Jill Johnson et al. The manuscript describes the use of a statistical emulator to explore the model parameter space in the HadGEM3–UKCA aerosol–climate model. It concludes that the parameter space that is consistent with synthetic "observations" of the present-day state of the climate system is large; that this large parameter space leads to a large spread in aerosol effective radiative forcing (ERF) estimates; and that other methods of estimat-

ing the ERF uncertainty, such as multi-model ensembles and "emergent constraints" applied to such ensembles, give a deceptively small answer.

The manuscript has a clear message and is well written (I particularly enjoyed the succinct discussion of the different meanings of "constraint" in the introduction), which made it very pleasant to read. The issue of reliable uncertainty estimates is an important one, and I believe the exploration of model parameter space is a key contribution toward making progress here. I therefore strongly support the publication of this manuscript. I have one main criticism that I believe needs to be addressed (Section 1). I also have a fairly minor suggestion to the authors which they should feel free to adopt or disregard (Section 2), as well as a litany of minor comments (Section 3).

**1  Are the synthetic observations used in this analysis representative of model tuning?**

The authors argue that producing one tuned configuration of a climate model underestimates the size of the parameter space (no disagreement there); that the tuning process is often done one parameter at a time, underestimating the non-additive effects of varying multiple parameters at once (only mild disagreement there); and that an analysis based on an ensemble of ERF estimates from tuned models underestimates the true ERF uncertainty. For this last conclusion to be true, the range of ERFs that a single model can plausibly produce ("plausible" meaning consistent with the observed present-day climate) must be non-negligible compared to the intermodel ERF spread, as sketched in Figure 10.

In their analysis, the authors subject the different parameter combinations of their emulated aerosol–climate model to a consistency check against nine observables of the present-day climate. This is the emulator-world equivalent of tuning a single climate model to agree with the present-day climate; the further conclusions made in the

manuscript about the uncertainty on the model's ERF estimate thus hinge crucially on whether this consistency with observations is indeed equivalent to model tuning.

I am concerned that the answer may be "no", for the following reason. The observational dataset used here is the July mean conditions over Europe. Europe is a tiny portion of the globe, and July only samples one point in the seasonal cycle of aerosol and cloud properties. Since, as the authors point out, the model uncertainty stems from the interaction of aerosol and host model parameters, it seems like one would want to use the widest spectrum of weather and aerosol conditions available to be able to discard observationally excluded parameter combinations.

As a GCM tuning strategy, a European seasonal approach would fail because the constraint on the global-mean climate would be negligible. I expect that many of the ensemble members that are consistent with the European July observations will have outlandish global-mean TOA fluxes, cloud fields, etc., that would get them rejected in a constraint that uses global observations. This, I expect, would narrow the estimate of the single-model uncertainty compared to what the authors find using their constraint strategy.

The interesting question, in my mind, is how much using global constraints would narrow the uncertainty, i.e., whether Figure 10(b) would still look mostly as it does now or start to look more like Figure 10(a).

I realize that Europe was chosen for a reason, namely that global observations do not exist for all of the fields used as constraints (CCN; decadal trends of surface shortwave radiation and AOD). The most accurate equivalence with GCM tuning would probably be to use global fields where they are available and European fields (but for more than a single month) otherwise.

**2 Some of the conclusions of the paper could be phrased as recommendations on the future direction of the aerosol forcing community**

Apart from the main finding of large single-model uncertainty, there are other interesting findings in the paper that I think are underemphasized:

- The emulator–PPE method is a necessary step toward estimating the model uncertainty correctly. But to narrow the uncertainty range, it will also be necessary to assemble the most powerful (i.e., discriminating) combination of observables. Therefore, I encourage the authors to advocate for both of these necessary steps at once – the more they go hand-in-hand, the more efficient we are likely to be at making progress. (For example, the emulator–PPE method could point to which missing observable would provide the greatest additional constraint if it were added to the existing set of observables; see my comment on Figure 9 in the next section.) This sentiment is kind of there in the last sentence of the abstract and in the text (p. 17, l. 20; p. 23, l. 16), but it would be a missed opportunity not to phrase it more forcefully and more positively in the abstract.

- The aerosol forcing change over the coming decades is a much easier target than the forcing relative to preindustrial conditions, assuming the emissions changes are known. Perhaps it is time to move away from ERF as the holy grail of the field and instead focus on future aerosol emissions scenarios.

- AOD is a terrible variable if your aim is to understand aerosol–cloud interactions. We should figure out something better. Ed Gryspeerdt's work seems to show that aerosol index is a much better proxy for CCN. If the authors' model diagnoses AI, it would be easy for them to refute or corroborate this result.

These results have a direct bearing on where the aerosol forcing community would best invest its efforts in order to reduce the forcing uncertainty. I understand that the authors

want to keep the focus on the main result (the large single-model uncertainty), but they might think about ways to give these other findings a prominent place as well. For example, in our recent review article on radiative forcing by aerosol–cloud interactions (https://doi.org/10.1007/s40641-018-0089-y), we eschewed the traditional re-listing of conclusions at the end of the paper in favor of making recommendations to the forcing community. Mentioning these recommendations in the abstract may be appropriate as well.

**3  Other minor comments**

- Table 1: A bit more detail on parameters 8 and 9 would be nice. (Threshold in what variable? Rate of change with respect to what?)

- p. 10, l. 8: Do we know that the validation carries over to this new emulator? What is the rationale for not simply using the emulator that is definitely validated?

- p. 10, l. 14: Do the two approaches (elicited vs uniform PDF) give very different results?

- p. 11, l. 10: This does not make intuitive sense to me; if the "truth" is chosen to lie at one the edge of parameter space, shouldn't ensemble members from the opposite direction of that edge be penalized much more strongly than they would be if the "truth" were chosen to lie in the center?

- p. 14, l. 2: Much as I hate linear correlation coefficients, perhaps it would be useful to tabulate them to make it easier to follow this discussion. I am having trouble reading them off Figure 2.

- Speaking of Figure 2:

- Nice to run into fellow R users.
- All the labels are tiny! `par(cex)` may help.
- Point clouds are hard to interpret. Perhaps the relationships between variables would be easier to interpret as color maps of the 2D densities of the emulator results?
- Raster graphics are an abomination. R supports PDF output; this is the best option in general, and in particular for an information-rich figure like this one, where the reader will want to zoom in on interesting features.
- These comments apply to Figure 3 as well, where additionally I see funny boxes around some of the panels at certain magnifications.

• Figure 4: In this model, ERF variability is dominated by ERFaci variability (see Figure 2). Yet the model does not care one bit about aerosol–precipitation interactions, at least not wet scavenging. I believe this is quite different from other GCMs. Any idea why? (Not necessarily something that needs to go into the paper.)

• p. 25, l. 15: In GCM tuning, this would routinely be done; see my main point of criticism.

• Figure 9: Does the converse also hold (that observations of decoupling would be a useful constraint on the model)?

• p. 27, l. 5: But Cherian et al. do this using models tuned to reproduce the global climate; see my main point of criticism.

• p. 28, l. 10: AOD multidecadal change appears to be double-counted.

• p. 29, l.1: Would AI or fine-mode fraction work better? I think the authors have the opportunity to make a significant statement here about whether there is a way

forward from AOD, which is known to be a poor CCN proxy, via other proxies. See my point in the recommendations section above.

- p. 31, l. 6: I don't understand the point about cancellation of correlated errors, but I would like to. Perhaps the authors could elaborate.

- Craig 1997 has a bunch of cryptic initials instead of editor names.

- Gryspeerdt 2017 a and b are the same publication.

- Stier ACPD 2015 has been superseded by Stier 2016 ACP (https://www.atmos-chem-phys.net/16/6595/2016/acp-16-6595-2016.html).

- Penner 2011: the DOI looks strange.

- Pujol 2008: check that this is still up to date with `citation("sensitivity")`.

- Zhang 2016: Toshi Takemura's name is misspelled.

I hope that these comments will be useful to the authors. I look forward to seeing their updated manuscript.

Best regards,

Johannes

---

## Referee Comment (RC2) · Anonymous Referee #2 · 21 Jun 2018

The paper is an important piece for the puzzle called "reducing uncertainty in aerosol forcing" and contains a lot of interesting investigations testing aerosol forcing uncertainty with multiple observations using an emulator. The paper should be published with minor changes.

p1 l29 "improvements in the physical realismn"….I dont think Mann et al 2014 is the right citation at exactly this point.

p2 l2: "although the set of models is different to those used to assess aerosol microphysical properties in Mann et al. (2014), " not really an argument for the stubborness of the ERF uncrtainty, can be omitted here.

[Figure]

P2 l17 I think this paragraph and equation is misleading in pretending "that the forcing depends on the interlinked sensitivities of aerosols, clouds and their radiative properties to changes in aerosol emissions ". Direct radiative effects, fast adjustments are not readily folded in into this equation. Please rephrase.

P3 l23: "there is no equivalent to Equation 1 defining how a bias in simulated aerosol properties affects the forcing " => I think this is overly critical to bias inspections. An underestimate in fine mode AOD or bias in absorption can be translated in forcing bias. Measurmentes of fine mode AOD estimates can constrain anthropogenic AOD to some extent. And there might be other clever interpretations of bias. Please rephrase.

P3 l31 "Model variants that produce implausible results are rejected and, likewise, the forcings that they calculate are also rejected. " => would be nice to explain this at this point a bit more. Do you look at all observations at the same time? What is the criterion for rejecting?

P5 l9 "The analysis is restricted to Europe for the month of July. We do this primarily because regional observations can provide a better constraint on model uncertainty than global mean observations .. but with the disadvantage of being less straightforward to understand. . . . We choose Europe because there are many long-term measurements " => I don't buy these arguments. With synthetic observations this should not be a big problem to do globally. There are no long term measurements used. I assume this is done to save computer time. I think its ok to use just Europe and just July. But the discussion should be more honest and open here. Paragraph please rewrite.

Chapter 2.1 and 2.2. and 2.3: I think they can be reversed. Some simple questions are not clear to me: Are the simulations global? Is it a one year simulation with a 4 month spinup (eg Sep-Dec of the preceding year) and is then just July analysed? Is the emulator producing global fields, from which data are sampled at European tations?

Page 5 counts 191 simulations, while page 9 counts "in total 217 perturbed parameter simulations". Better to harmonize numbers.

Conclusions: I wonder how general the findings are if the ERF is in essence tested only over Europe and July with synthetic observations, but that might be shown in future publications.

---

## Referee Comment (RC3) · Anonymous Referee #3 · 26 Jun 2018

First of all I think this is a very useful and important work. Nicely show how this results will be important for emergent constrain multi model studies, and shows how different observational constraints individually can affect the ERF uncertainties.

One comment I have, is that it should be made more clear in abstract and conclusion, and also some figures and tables, that they use synthetic observations and not real observations. And define synthetic observations the first time it is mentioned.

Page 7 line 10. Specify that it is biomass burning emissions.

Table 2: Indicate that this is not real observations. Useful to define Europe also. In addition to the synthetic observations, real observations are used for ToA flux, am I

[Figure]

right?

Figure 9: Does the color shading mean anything? Include a colorbar or remove the shading.

[Figure]

---

## Author Comment (AC1) · 6 Aug 2018

In our response, reviewer comments are marked in bold, our responses and original text in plain text, and altered text in the paper in bold italic.

**Response to reviewer 1 (J. Mülmenstädt)**

We thank the reviewer for their interesting and useful comments on our manuscript. Our responses to these comments are given below.

**Major comment (Section 1): Are the synthetic observations used in this analysis representative of model tuning?**

**The authors argue that producing one tuned configuration of a climate model underestimates the size of the parameter space (no disagreement there); that the tuning process is often done one parameter at a time, underestimating the non-additive effects of varying multiple parameters at once (only mild disagreement there); and that an analysis based on an ensemble of ERF estimates from tuned models underestimates the true ERF uncertainty. For this last conclusion to be true, the range of ERFs that a single model can plausibly produce ("plausible" meaning consistent with the observed present-day climate) must be non-negligible compared to the intermodel ERF spread, as sketched in Figure 10.**

**In their analysis, the authors subject the different parameter combinations of their emulated aerosol–climate model to a consistency check against nine observables of the present-day climate. This is the emulator-world equivalent of tuning a single climate model to agree with the present-day climate; the further conclusions made in the manuscript about the uncertainty on the model's ERF estimate thus hinge crucially on whether this consistency with observations is indeed equivalent to model tuning.**

**I am concerned that the answer may be "no", for the following reason. The observational dataset used here is the July mean conditions over Europe. Europe is a tiny portion of the globe, and July only samples one point in the seasonal cycle of aerosol and cloud properties. Since, as the authors point out, the model uncertainty stems from the interaction of aerosol and host model parameters, it seems like one would want to use the widest spectrum of weather and aerosol conditions available to be able to discard observationally excluded parameter combinations.**

**As a GCM tuning strategy, a European seasonal approach would fail because the constraint on the global-mean climate would be negligible. I expect that many of the ensemble members that are consistent with the European July observations will have outlandish global-mean TOA fluxes, cloud fields, etc., that would get them rejected in a constraint that uses global observations. This, I expect, would narrow the estimate of the single-model uncertainty compared to what the authors find using their constraint strategy.**

**The interesting question, in my mind, is how much using global constraints would narrow the uncertainty, i.e., whether Figure 10(b) would still look mostly as it does now or start to look more like Figure 10(a).**

**I realize that Europe was chosen for a reason, namely that global observations do not exist for all of the fields used as constraints (CCN; decadal trends of surface shortwave radiation and AOD). The most accurate equivalence with GCM tuning would probably be to use global fields where they are available and European fields (but for more than a single month) otherwise.**

We don't tackle the problem of global mean ERF or global tuning; the paper is focused solely on Europe. The word tuning doesn't need to imply global. Aerosol models are frequently tuned or adjusted in some way (call it what you will) to agree with observations in a particular region or environment, so it's this process we are drawing comparisons with. We restrict the analysis to Europe because there is a distinct set of parameters that are important to aerosol properties and ERF uncertainty in each region (see Lee et al, 2016: 10.1073/pnas.1507050113 for a map of clusters of uncertain parameters). Because parameter sensitivities vary regionally, to constrain the global forcing you would need to constrain the different regions individually – i.e., in the aerosol world it doesn't make much sense to talk about global tuning. So our European study is like a mini version of what one would need to do globally. If it doesn't work for Europe (with a distinct set of parameters) then it won't work globally either. Using global ToA flux would have limited effect because of many regional compensating factors. We tested this in Regayre et al, 2018 (10.5194/acp-18-9975-2018; section 3.5).

Another reason to use Europe was to relate to the Cherian et al, 2014 study where European surface solar radiation trends were used as an emergent constraint on regional and global forcing. As we state in our conclusions, we think the constraint will be weaker than they calculated because of unconstrainable uncertainties in the individual models.

So to answer the question "how much using global constraints would narrow the uncertainty", the answer is that it would narrow the *global* ERF more than we would currently achieve. But a global constraint won't have much effect on Europe (again, because of the distinct set of parameter uncertainties). We only ever refer to Europe, and we maintain that focus when we compare with the Cherian et al Europe SSR constraint on Europe-mean ERF in Fig 10.

**Minor suggestion (Section 2): Some of the conclusions of the paper could be phrased as recommendations on the future direction of the aerosol forcing community**

This is an interesting suggestion and we are pleased that the reviewer has identified the potential impacts that our work and results could have in the aerosol forcing community. We did try to draw out the implications. We prefer not to completely change the style of the conclusions because we think it's important that we include a synthesis of findings and integration with current knowledge, rather than recommending future studies. We think recommendations are more appropriate for perspective articles and reviews. In response to some specific suggestions we have tried to highlight the recommendations a bit.

**Apart from the main finding of large single-model uncertainty, there are other interesting findings in the paper that I think are underemphasized:**

- **The emulator–PPE method is a necessary step toward estimating the model uncertainty correctly. But to narrow the uncertainty range, it will also be necessary to assemble the most powerful (i.e., discriminating) combination of observables. Therefore, I encourage the authors to advocate for both of these necessary steps at once – the more they go hand-in-hand, the more efficient we are likely to be at making progress. (For example, the emulator–PPE method could point to which missing observable would provide the greatest additional constraint if it were added to the existing set of observables; see my comment on Figure 9 in the next section.) This sentiment is kind of there in the last sentence of the abstract and in the text (p. 17, l. 20; p. 23, l. 16), but it would be a missed opportunity not to phrase it more forcefully and more positively in the abstract.**

We have added a sentence on this to the end of abstract to highlight this point. The end of the abstract now reads:

"…However, the uncertainty in the aerosol ERF after observational constraint is large compared to the typical spread of a multi-model ensemble. Our results therefore raise questions about whether the underlying multi-model uncertainty would be larger if similar approaches as adopted here were applied more widely. ***The approach presented in this study could be used to identify the most effective observations for model constraint.*** It is hoped that aerosol ERF uncertainty can be further reduced by introducing process-related constraints, however, any such results will be robust only if the enormous number of potential model variants is explored."

- **The aerosol forcing change over the coming decades is a much easier target than the forcing relative to preindustrial conditions, assuming the emissions changes are known. Perhaps it is time to move away from ERF as the holy grail of the field and instead focus on future aerosol emissions scenarios.**

We agree, and have made this point in at least a couple of previous papers. There is a paragraph (5[th] of the conclusions in the original manuscript; 6[th] paragraph in the revised manuscript) dealing with this point, but we agree it could be focused. We now write:

"Observational constraint using nine observations has the potential to reduce the uncertainty in aerosol ERF ***slightly more over a multi-decadal period than over the full industrial period***:"

The rest of the paragraph then explains why this is important. Then we add at the end of the paragraph:

"***A shift of emphasis of the research community towards trying to constrain decadal forcing uncertainty, instead of industrial era forcing, is likely to accelerate progress.***"

- **AOD is a terrible variable if your aim is to understand aerosol–cloud interactions. We should figure out something better. Ed Gryspeerdt's work seems to show that aerosol index is a much better proxy for CCN. If the authors' model diagnoses AI, it would be easy for them to refute or corroborate this result.**

Unfortunately, we cannot compute AI from the model runs of this PPE and so we cannot evaluate this here. However, we disagree that AOD is unambiguously a "terrible variable" for constraining forcing. This point of view is maybe based on the fact that AOD and CCN are not closely related but AI and CCN are more related. However, our study shows that neither CCN nor AOD alone provides a strong constraint on ERF (Fig 7). In that sense you might argue that CCN is also a terrible variable. The key for model constraint and uncertainty reduction is to find *combinations of observables* that are sensitive to the same set of process or emission uncertainties as ERF. So *all* observed variables are helpful in their own way. It is possible that AI will be useful for direct extrapolation back to the pre-industrial period, but it will not necessarily be useful for model constraint. We intend to include AI in the output diagnostics of a future PPE that is in the planning stage, with which we will be able to investigate the potential of AI for model constraint.

We have added a new paragraph in the conclusions (after the 4[th] paragraph in the original manuscript):

"***It is often argued that AOD is a poor variable to use for understanding aerosol-cloud interactions. However, our results show that even the most strongly related measurement (CCN) also does not***

*provide a strong individual constraint on ERF$_{ACI}$ (Figure 7). It is doubtful that other derived variables like aerosol index will be any better. The key to model constraint is to find combinations of observations that help to constrain ERF: Individual constraints are unlikely to be effective, although they may appear to be effective if the model uncertainty is not fully sampled.*"

**These results have a direct bearing on where the aerosol forcing community would best invest its efforts in order to reduce the forcing uncertainty. I understand that the authors want to keep the focus on the main result (the large single-model uncertainty), but they might think about ways to give these other findings a prominent place as well. For example, in our recent review article on radiative forcing by aerosol–cloud interactions (https://doi.org/10.1007/s40641-018-0089-y), we eschewed the traditional re-listing of conclusions at the end of the paper in favor of making recommendations to the forcing community. Mentioning these recommendations in the abstract may be appropriate as well.**

We understand how useful a set of recommendations to the aerosol forcing community would be, but we have decided not to fully re-work our conclusions in this paper to a set of recommendations. The presented study, based only on synthetic observations, corresponds to a specific stage in our overall work to understand and constrain the aerosol forcing uncertainty in this aerosol-climate model. Our current and future work is now expanding this work to use real observations in the constraint methodology, and we are learning even more as we move forward with this approach. We feel that recommendations to the community must take into account findings from the complete constraint process in which real observations are used. We therefore elect to defer producing such a set of recommendations at this earlier stage to a future time when we also have a comprehensive evaluation of model constraint with real observations.

**Minor Comment 1: Table 1: A bit more detail on parameters 8 and 9 would be nice. (Threshold in what variable? Rate of change with respect to what?)**

Full descriptions of the parameters perturbed in this PPE are given in Yoshioka et al, 2018 and Regayre et al, 2018. We therefore do not repeat this full detail in this publication.  We have made it clearer to the reader where more detailed descriptions of the parameters can be found by moving the reference for this from the end of paragraph 3 (page 8 line 25 in the original manuscript) to paragraph 2 in Section 2.3. Paragraph 2 of Section 2.3 now reads:

"The 27 perturbed parameters are listed in Table 1. They are categorized as either aerosol (aer) or atmospheric (atm) according to their role in the model. To define the set of parameters we used expert elicitation and carried out one-at-a-time parameter perturbation screening experiments to quantify the effect of individual parameter perturbations away from the default setting*. **The selected parameters are described in more detail in related papers (Regayre et al., 2018; Yoshioka et al., 2018)**"

For the specific parameters mentioned here, we have adapted the text in Table 1 to provide further clarity.

For parameter 8: Dec_Thres_Cld, we have changed the description to: "***Threshold for the ratio of buoyancy consumption to production before decoupling occurs***"

For parameter 9: Fac_Qsat, we have updated the description to: "Rate of change in convective parcel maximum condensate **with altitude**"

**Minor Comment 2: p. 10, l. 8: Do we know that the validation carries over to this new emulator? What is the rationale for not simply using the emulator that is definitely validated?**

We are confident that the validation does carry over to the new emulator built using all simulations. The rationale for doing this is to incorporate all of the information that we have from our PPE into the emulator model we use.

We do investigate the validity of the final emulator using a process called 'leave-one-out' validation on the merged set of simulations. In this procedure we remove a simulation from the set, build a new emulator and then use that emulator to predict the removed simulation, repeating this for all simulations in the set in turn. This procedure has returned good validation prediction results in all cases, and Figure A below shows an example of this validation for our present-day AOD emulator. The plot shows that the real model output and corresponding Leave-one-out emulator predictions (with 95% credible intervals) strongly correlate along a line of equality. Hence, this alternate validation indicates that the addition of the extra simulations in the construction of the emulator does not deteriorate the emulator quality and we have confidence in this approach.

[Figure]

Figure A: Example output of a 'Leave-one-out' validation test of the final emulator.

We have added the following sentence to page 10, line 9 of the original manuscript (also page 10, line 9 of the revised manuscript) to detail that we use a leave-one-out validation procedure to verify the quality of the final emulator:

"***A 'leave-one-out' validation procedure (where each simulation in turn is removed from the merged set, and a new emulator is constructed and used to predict that removed simulation) is applied to additionally verify the quality of our final emulator.***"

Further explanation on our rationale for this is given below for the reviewer's interest.

Given a good validation of an initial emulator based only on the original training runs, we could certainly use this emulator for our analysis and have confidence in the results. However, the validation simulations are purposely placed into gaps between the locations of the training simulations and so including this extra information enables us to better guide/tweak the predicted

emulator response in these gaps to be even more in-line with the underlying 'true' model output. We believe that if a good emulator is achieved based on only the training runs, then a further emulator based on both the training and validation runs must be of similar quality, if not better, given the inclusion of the extra information that these extra runs provide.

**Minor Comment 3: p. 10, l. 14: Do the two approaches (elicited vs uniform PDF) give very different results?**

In general, we have found that the overall constraint achieved on regional average model outputs (aerosol properties and forcings) is of a very similar nature under both approaches (using the elicited v's using the uniform pdfs). However, there are differences in how the generated large sample of model variants (size 4 million) covers the multi-dimensional parameter uncertainty and hence the proportion of that sample that is retained on application of the constraint methodology.

Using elicited pdfs rather than uniform pdfs when sampling over the multi-dimensional parameter uncertainty space essentially applies a further constraint to that parameter space, sampling much less in areas that the experts believe to be highly unlikely. This means that the sample of model variants tested against the observations in the constraint procedure is more concentrated towards the most likely (as evaluated by our experts) parameter values in the parameter ranges. The areas sampled less tend to be around the edges of the multi-dimensional space.

On constraint we have found that using the elicited pdfs means we retain more of the 4 million model variants and leads us to a slightly stronger constraint on the parameter space itself (e.g. on the marginal parameter space in Figure 9). We reject more variants using uniform pdfs as the corresponding sample under this assumption covers more densely the parts of the parameter space that were deemed unlikely to be plausible by the experts in the elicited pdfs. This result shows some consistency between the model behaviour and the expert judgements, which gives us confidence in using the elicited pdfs. Hence, in the results presented we choose to use the elicited pdfs and so include all available information in the sampling in order to determine the full potential of possible constraint on ERF that is achievable with our approach.

**Minor Comment 4: p. 11, l. 10: This does not make intuitive sense to me; if the "truth" is chosen to lie at one the edge of parameter space, shouldn't ensemble members from the opposite direction of that edge be penalized much more strongly than they would be if the "truth" were chosen to lie in the center?**

Yes, it is correct that choosing the "truth" to lie in a different part of the parameter space can affect the corresponding 'observed' output values and so change the model variants (and hence parts of parameter space) that are retained on constraint. However, the effect of this change on model output quantities like aerosol forcing is dependent on the way in which the response surfaces of these outputs vary over the multi-dimensional parameter space, and so it is difficult to predict. The compensating effects of the different parameters in the multi-dimensional space can mean that different areas of parameter space lead to similar output values.

For the model run we investigated as a marginal set of synthetic observations (our validation run 27, which had several parameter values towards the edges of their uncertainty range), we found that the achieved overall constraint on each of the forcing variables (using all observations together) was slightly greater and we also retained fewer model variants in the constrained sample. For ERF we

saw a further reduction in the standard deviation of around 12%, however this reduction was much lower for $ERF_{ARI}$ at only 1%. Even so, the relative individual constraint effect of each observational type (Figure 7) was of the same nature/order for all four forcing variables (ToA flux is the most effective for ERF and $ERF_{ACI}$, the sulphate concentration is most effective for $ERF_{ARI}$ and $ERF_{ARIclr}$, and in general the achieved constraint from the individual observable aerosol properties is weak). Hence, the overall conclusions of the study did not change on using a different observation set. The resulting constrained set of model variants still corresponds to a large multi-dimensional area of parameter space with a lot of equifinality.

Given these results, and so as not to over-complicate the presented study, we made the decision to focus on just one observation case and the centralised set (with parameters set to their elicited median values) seemed most appropriate for a general example of our approach.

**Minor Comment 5: p. 14, l. 2: Much as I hate linear correlation coefficients, perhaps it would be useful to tabulate them to make it easier to follow this discussion. I am having trouble reading them off Figure 2.**

We have added the following table of linear correlation coefficients into Section 3.1 of the manuscript:

| ToA Flux | ΔSSR | CCN | AOD | ΔAOD | $PM_{2.5}$ | Sulphate | OC | BC | ERF | $ERF_{ACI}$ | $ERF_{ARI}$ | $ERF_{ARIclr}$ |
|---|---|---|---|---|---|---|---|---|---|---|---|---|
| **ToA Flux** | 0.20 | 0.00 | 0.23 | -0.30 | 0.16 | 0.10 | 0.04 | -0.03 | -0.59 | -0.65 | 0.25 | -0.05 |
| | **ΔSSR** | -0.04 | 0.22 | -0.72 | 0.12 | 0.49 | -0.04 | 0.11 | -0.32 | -0.22 | -0.52 | -0.61 |
| | | **CCN** | 0.46 | -0.20 | 0.21 | 0.03 | 0.37 | 0.09 | -0.14 | -0.14 | -0.01 | -0.09 |
| | | | **AOD** | -0.50 | 0.88 | 0.59 | 0.66 | 0.59 | -0.33 | -0.27 | -0.33 | -0.55 |
| | | | | **ΔAOD** | -0.32 | -0.71 | -0.17 | -0.20 | 0.48 | 0.38 | 0.55 | 0.76 |
| | | | | | **$PM_{2.5}$** | 0.54 | 0.69 | 0.54 | -0.21 | -0.15 | -0.33 | -0.44 |
| | | | | | | **Sulphate** | 0.47 | 0.64 | -0.41 | -0.28 | -0.67 | -0.77 |
| | | | | | | | **OC** | 0.62 | -0.34 | -0.31 | -0.20 | -0.30 |
| | | | | | | | | **BC** | -0.27 | -0.22 | -0.28 | -0.46 |
| | | | | | | | | | **ERF** | 0.98 | 0.16 | 0.48 |
| | | | | | | | | | | **$ERF_{ACI}$** | -0.04 | 0.33 |
| | | | | | | | | | | | **$ERF_{ARI}$** | 0.83 |
| | | | | | | | | | | | | **$ERF_{ARIclr}$** |

**Table 3:** Pearson linear correlations (r) between the PPE member regional mean model outputs for Europe in July, for the aerosol properties used as constraints, corresponding to the pairwise scatter plots in Figure 2.

We have edited the first paragraph of Section 3.1 to say:

"Figure 2 shows pairwise scatter plots of the PPE member output (Europe July-mean), which provides an overview of the spread of the model outputs as well as the relationships between the variables. *We further quantify any linear relationships between the variables using the Pearson correlation coefficient (r) in Table 3.*"

We have adjusted the numbering of all following results tables in the manuscript to incorporate this table into Section 3.1.

**Minor Comment 6: Speaking of Figure 2:**

- **Nice to run into fellow R users.**
- **All the labels are tiny! par(cex) may help.**

It is difficult with so many small plots in the same figure to make the labels any larger. We have used 'par(cex)' to adjust the size of the axis labels to gain the best possible labelling size whilst avoiding axis values overlapping and becoming unreadable.

- **Point clouds are hard to interpret. Perhaps the relationships between variables would be easier to interpret as color maps of the 2D densities of the emulator results?**

The purpose of this figure is to look at the direct model output from the 191 individual model runs before any interpolation over the parameter uncertainty space (using the emulators) is applied. With only 191 points in each case, we do not have enough data to generate robust density contours and therefore use scatter plots.

- **Raster graphics are an abomination. R supports PDF output; this is the best option in general, and in particular for an information-rich figure like this one, where the reader will want to zoom in on interesting features.**

Figure 2 has been re-made using postscript and PDF graphics, and the new version is much improved for zooming in on interesting features. The new Figure 2 is below (at the end of this comment response)

**These comments apply to Figure 3 as well, where additionally I see funny boxes around some of the panels at certain magnifications.**

We have looked in detail at Figure 3 and cannot find any issues with the figure on magnification. We therefore have not changed this figure. We will check the proofs.

[Figure]

**Figure 1.** Pairwise scatter plots of the PPE member regional mean model output for Europe in July, for the aerosol properties used as constraints: ToA flux (W m$^{-2}$), change in SSR ($\Delta$SSR, W m$^{-2}$) between 1978 and 2008, CCN conc. (cm-3), AOD, surface mass concentrations of **PM$_{2.5}$**, Sulphate, OC, and BC ($\mu$g m$^{-3}$), the changes in AOD ($\Delta$AOD, W m$^{-2}$) between 1978 and 2008, and the 1850-2008 forcing variables: aerosol ERF, ERF$_{ACI}$, ERF$_{ARI}$ and ERF$_{ARIclr}$ (W m$^{-2}$).

**Minor Comment 7: Figure 4: In this model, ERF variability is dominated by ERFaci variability (see Figure 2). Yet the model does not care one bit about aerosol–precipitation interactions, at least not wet scavenging. I believe this is quite different from other GCMs. Any idea why? (Not necessarily something that needs to go into the paper.)**

The cloud state is sensitive to precipitation scavenging, but the ERF$_{ACI}$ is not. The ERF is obviously a change from pre-industrial to present-day, so it is not obvious that if you change the aerosol/cloud state in both periods that the ERF will remain sensitive to scavenging.

**Minor Comment 8: p. 25, l. 15: In GCM tuning, this would routinely be done; see my main point of criticism.**

OK.

**Minor Comment 9: Figure 9: Does the converse also hold (that observations of decoupling would be a useful constraint on the model)?**

We do not know what the reviewer means by "observations of decoupling", and so we are not sure on what is being referred to here.  We are therefore unable to comment.

**Minor Comment 10: p. 27, l. 5: But Cherian et al. do this using models tuned to reproduce the global climate; see my main point of criticism.**

Yes, we agree that Cherian et al, 2014 achieve their constraint using a set of models that have each been 'tuned' in some way to re-produce the global climate. We compare with our 'tuned' model for Europe (which includes the frequently-tuned quantity, the ToA flux). However, in the selection of each individual tuned model, the influence of parametric uncertainty has not be rigorously explored. This means that there will be many other equally plausible tuned versions of each given model that agree with the observations and could have been selected, as we have shown in this study for the single tuned model HadGEM3-UKCA. We have shown that the predicted forcing from the set of the equally plausible model variants (obtained through comparison to a diverse set of aerosol observations) is wide-ranging. This implies that tuning does not necessarily directly reduce the model spread and implies that emergent constraints such as the one obtained by Cherian et al, 2014 likely underestimate the true spread of forcing.

**Minor Comment 11: p. 28, l. 10: AOD multidecadal change appears to be double-counted.**

This has been corrected in the manuscript. This sentence in the first paragraph of the conclusions (page 30 lines 5-9 of the revised manuscript) now reads:

"The primary objective of our study was to determine how much uncertainty could remain in an aerosol-climate model when it is constrained to match combinations of observations that define the base state of the model: top-of-atmosphere upward shortwave flux, aerosol optical depth, PM2.5, cloud condensation nuclei, concentrations of sulphate, black carbon and organic material as well as multi-decadal change in surface shortwave radiation and aerosol optical depth."

**Minor Comment 12: p. 29, l.1: Would AI or fine-mode fraction work better? I think the authors have the opportunity to make a significant statement here about whether there is a way forward from AOD, which is known to be a poor CCN proxy, via other proxies. See my point in the recommendations section above.**

Please see our response to the Minor suggestion (Section 2) above (at the end of page 3). We are not able to evaluate the effects of AI with this PPE, but we doubt that AI would provide any better constraint on the aerosol ERF than AOD. AI might be a better extrapolation variable, but this has not yet been directly demonstrated.

**Minor Comment 13: p. 31, l. 6: I don't understand the point about cancellation of correlated errors, but I would like to. Perhaps the authors could elaborate.**

We mean that a model might have a large bias in AOD and in CCN (so neither can be simulated well) but if both of these model variables are biased for the same reason (e.g., incorrect aerosol deposition rates), then the ratio of CCN/AOD might be accurately simulated by the model.

**Minor Comment 14: Craig 1997 has a bunch of cryptic initials instead of editor names.**

This reference has been corrected in the manuscript.

**Minor Comment 15: Gryspeerdt 2017 a and b are the same publication.**

This has been corrected in the manuscript.

**Minor Comment 16: Stier ACPD 2015 has been superseded by Stier 2016 ACP (https://www.atmos-chem-phys.net/16/6595/2016/acp-16-6595-2016.html).**

This reference has been corrected in the manuscript.

**Minor Comment 17: Penner 2011: the DOI looks strange.**

This reference has been corrected in the manuscript.

**Minor Comment 18: Pujol 2008: check that this is still up to date with citation("sensitivity").**

This reference has been checked in R and corrected in the manuscript.

**Minor Comment 19: Zhang 2016: Toshi Takemura's name is misspelled.**

This reference has been corrected in the manuscript.

**References:**

Cherian, R., Quaas, J., Salzmann, M. and Wild, M.: Pollution trends over Europe constrain global aerosol forcing as simulated 25 by climate models, Geophys. Res. Lett., 41(6), 2176–2181, doi:10.1002/2013GL058715, 2014.

Lee, L. A., Reddington, C. L. and Carslaw, K. S.: On the relationship between aerosol model uncertainty and radiative forcing uncertainty., Proc. Natl. Acad. Sci. U. S. A., 113(21), 5820–7, doi:10.1073/pnas.1507050113, 2016.

Regayre, L. A., Johnson, J. S., Yoshioka, M., Pringle, K. J., Sexton, D. M. H., Booth, B. B. B., Lee, L. A., Bellouin, N., and Carslaw, K. S.: Aerosol and physical atmosphere model parameters are both important sources of uncertainty in aerosol ERF, Atmos. Chem. Phys., 18, 9975-10006, doi:10.5194/acp-18-9975-2018, 2018.

Yoshioka, M., Regayre, L., Pringle, K. J., Mann, G. W., Sexton, D. M. H., Johnson, C. E. and Carslaw, K. S.: Perturbed parameter ensembles of the HadGEM-UKCA composition-climate model to explore aerosol and radiative forcing uncertainty, J. Adv. Model Earth Syst., in-prep, 2018.

---

## Author Comment (AC2) · 6 Aug 2018

In our response, reviewer comments are marked in bold, our responses and original text in plain text, and altered text in the paper in bold italic.

**Response to reviewer 2 (Anonymous reviewer)**

We thank the reviewer for their interesting and useful comments on our manuscript. Our responses to these comments are given below.

**Minor Comment 1: p1 l29 "improvements in the physical realism"... I don't think Mann et al 2014 is the right citation at exactly this point.**

We have changed this to "Although extensive improvements in the physical realism of aerosol-climate models have been made in recent years (Ghan and Schwartz, 2007*), resulting in a set of quite sophisticated models (*Mann et al., 2014)." Mann et al, 2014 is really the only reference for where a large set of microphysics models was assembled.

**Minor Comment 2: p2 l2: "although the set of models is different to those used to assess aerosol microphysical properties in Mann et al. (2014)," not really an argument for the stubbornness of the ERF uncertainty, can be omitted here.**

We have deleted that sentence. It wasn't really an argument, but really just a reminder that we should not compare these two inter-comparisons (Mann et al, 2014 microphysics models and Boucher et al, 2013 climate models) – they are completely different models.

**Minor Comment 3: P2 l17 I think this paragraph and equation is misleading in pretending "that the forcing depends on the interlinked sensitivities of aerosols, clouds and their radiative properties to changes in aerosol emissions". Direct radiative effects, fast adjustments are not readily folded in into this equation. Please rephrase.**

We are referring only to the aerosol-cloud forcing here. This equation is not pretending anything; it is the community's main approach to understanding how aerosol emission changes affect cloud properties. We have re-written the start of this paragraph (3$^{rd}$ paragraph in the introduction) on page 2 line 14 to clarify this applies only to aerosol-cloud forcing:

"There are three ways in which observations help to constrain the uncertainty in aerosol ERF. The first, *which applies to the aerosol-cloud-related forcing,* is based on..."

**Minor Comment 4: P3 l23: "there is no equivalent to Equation 1 defining how a bias in simulated aerosol properties affects the forcing " => I think this is overly critical to bias inspections. An underestimate in fine mode AOD or bias in absorption can be translated in forcing bias. Measurements of fine mode AOD estimates can constrain anthropogenic AOD to some extent. And there might be other clever interpretations of bias. Please rephrase.**

We disagree. In fact this is one of the main results of our paper: we show that aerosol-radiation interaction forcing (direct effect) is not strongly constrained by state variable measurements (AOD, etc.). There are many ways in which a model can be configured to get a particular AOD, but these

model variants (as we call them) predict very different forcings. To make this clear, and to signpost the result, we add at this point:

"(i.e., there is no equivalent to Equation 1 defining how a bias in simulated aerosol properties affects the forcing). One aim of our study is to make that link*, and we show in section 3.5.1 that observational constraint of many state variables only weakly constrains the direct and indirect radiative forcings.*"

**Minor Comment 5: P3 l31 "Model variants that produce implausible results are rejected and, likewise, the forcings that they calculate are also rejected. " => would be nice to explain this at this point a bit more. Do you look at all observations at the same time? What is the criterion for rejecting?**

The sentence referred to here is in the introduction section where we very briefly summarise / introduce the approach we take in this study. We therefore don't want to go into too much detail, as full details of the methodology and constraint approach are given in the following methodology section. Hence, we have only added very brief extra explanations of these points in this introductory paragraph/section of the manuscript.

In the paragraph on page 3, line 31 of the original manuscript (page 3, line 32 of the revised manuscript), we have added:

"... Model variants that produce implausible results *(i.e., output outside of an observation's estimated uncertainty range)* are rejected and, likewise, the forcings that they calculate are also rejected. A similar constraint methodology has been applied to..."

And we have added the following sentence to the end of this paragraph on page 4 line 3 of the original manuscript (page 4, line 4 of the revised manuscript):

"*We constrain using each aerosol/cloud observation individually and combinations of all observations.*"

We have then added more detail on our criteria for retaining/rejecting model variants in the constraint process in Section 2.7 (Identification of plausible model variants) of the methodology section, to make this process clearer. The start of Section 2.7 now reads as:

"Observationally plausible model variants are defined to be those that simulate aerosol and radiation properties within the uncertainty ranges of the observations, defined in Table 2. *As we use statistical emulators to generate the simulated output values for each model variant, rather than using the climate model directly, an emulator prediction error φ (valued at one standard deviation on the emulator prediction from the Gaussian process uncertainty) is also taken into account. Hence, for a given observed variable, a model variant is rejected as implausible if the range defined by its emulator prediction +/- φ lies outside the corresponding observation's uncertainty range in Table 2. Furthermore, for a joint observational constraint we retain only the model variants that are classed as plausible for all individual observation types that make up the joint constraint.*"

**Minor Comment 6: P5 l9 "The analysis is restricted to Europe for the month of July. We do this primarily because regional observations can provide a better constraint on model uncertainty than global mean observations... but with the disadvantage of being less straightforward to understand. . . . We choose Europe because there are many long-term measurements" => I don't buy these arguments. With synthetic observations this should not be a big problem to do globally. There are no long term measurements used. I assume this is done to save computer time. I think its ok to use just Europe and just July. But the discussion should be more honest and open here. Paragraph please rewrite.**

There was no intension on our part to not be honest and open in terms of our arguments for only using observations over the Europe region in July for this study.

Our full reasoning to base the presented study on only Europe in July is as follows:

- Previous work (Regayre et al, 2018, for example) has shown that using global mean quantities for constraint can mask many compensating regional parameter effects, leading to a very weak 'watered down' constraint on both the parameter space and model outputs like forcing that can be difficult to interpret.

- To constrain forcing globally we need to constrain the parameters that affect the forcing across the globe. Regayre et al, 2018 show that different parameter sources control the uncertainty in aerosols and forcing in different regions. Therefore, the global problem essentially breaks down to be the sum of constraining the forcing in a set of key regions, of which Europe is one. The Europe region in July provides a single region/month for which a distinct set of parameter uncertainties affect ERF. If we cannot constrain the forcing regionally in Europe, then we are unlikely to obtain a constraint on a global/multi-region scale. Hence, Europe in July provides us with the full insight we aim for here on the potential of our approach.

- It is true that our analysis is highly computational and generates a significant amount of data. We have investigated other regions in this work, including China and the North Pacific (not shown), but including more regions in the presented study would only significantly expand the results in terms of quantity and complexity, with no real gain as to our actual aim of establishing the overall potential for constraint with our methodology.

- Real observations of multiple aerosol observable quantities are sparse in many regions around the globe, and temporally (Reddington *et al.*, 2017), but Europe is a region for which a diverse set of aerosol observations are available. These observations provide realistic estimates of observational uncertainty for the synthetic study.

- The presented study was a specific stage in our model evaluation work, at which we aimed to test our methodology for constraint using synthetic observations before moving forward to our now current work where we are looking at using real observations. Using Europe in July for our synthetic study supplies a good test for evaluating the potential constraint we may achieve from using the real observations in the future – a test that we are now working towards verifying.

We have edited the penultimate paragraph in the introduction section at page 5 line 9 of the original manuscript (page 5 line 11 of the revised manuscript) to better reflect these reasons (The start of this paragraph also contains revisions with respect to our response to Reviewer 3's minor comment 2):

"The analysis is restricted to the region of Europe (defined in this study by the longitude range: 12°W to 41°E, and latitude range: 37.5°N to 71.5°N) for the month of July. We take a regional approach primarily because regional observations provide a better constraint on model uncertainty than global mean observations (Regayre et al., 2018). The sources of uncertainty in aerosols and forcing vary regionally (Lee et al., 2016; Reddington et al., 2017; Regayre et al., 2015). *Therefore,* a global analysis would essentially be a scaled-up version of what we present here – i.e., a set of *regional evaluations.* We choose Europe *in July as this is a region and month for which a distinct set of parameter uncertainties affect the aerosol properties and the ERF, providing a good test case for our methodology. Europe is also a region for which a diverse set of* long-term measurements of different aerosol and radiative properties *are* available, *that* we can use to inform our assessments of the observational uncertainty."

**Minor Comment 7: Chapter 2.1 and 2.2 and 2.3: I think they can be reversed. Some simple questions are not clear to me: Are the simulations global? Is it a one year simulation with a 4 month spinup (eg Sep-Dec of the preceding year) and is then just July analysed? Is the emulator producing global fields, from which data are sampled at European stations?**

We have considered the reviewers suggestion to change the order of the sub-sections in our methodology section (Section2) of the manuscript. However, we think that the current order is most suitable, as we prefer to keep the overall summary of our approach at the start (section 2.1), with the different aspects further explained in the order they are mentioned in that summary.

We have clarified the model set up in the final paragraph of section 2.3 (page 9 lines 11-12 of the original manuscript; page 9 lines 15-17 in the revised manuscript):

"... In total 217 perturbed parameter simulations *of the global model* were run *for a full year* for each anthropogenic emission period (1850, 1978 and 2008 emissions). *Each simulation had a spin-up period of seven months from a consistent starting simulation, where the parameters were set to their median values for the first four months and the perturbations then applied in the final three months*."

The emulators do not produce global fields. We have clarified this at the beginning of Section 2.4 (page 10, line 6 of the original manuscript; page 10 line 5 of the revised manuscript):

"For each model output (such as *the regional mean ToA flux, CCN conc., etc. for Europe in July*) we construct a statistical emulator model over the 27-dimensional parameter uncertainty..."

**Minor Comment 8: Page 5 counts 191 simulations, while page 9 counts "in total 217 perturbed parameter simulations". Better to harmonize numbers.**

The reasoning for this difference is explained in the next sentence on page 9 (lines 12-13). We only use ensemble members that completed the full year of simulation in all periods, which reduces the number of runs used for analysis to 191 from the total of 217 that were originally run. We feel it is important that we are transparent about this and so we continue to state both numbers. We have amended the sentence at page 9 line 12 of the original manuscript (page 9 line 18 of the revised manuscript) to improve the clarity on this point:

"Twenty-five simulations did not complete **the full** annual cycle **so were not used in our analysis. Consequently,** the ensemble of simulations **used for analysis** for each period was made up of the remaining 191 simulations, all of which were used to build the final emulators."

**Conclusions: I wonder how general the findings are if the ERF is in essence tested only over Europe and July with synthetic observations, but that might be shown in future publications.**

The aim of this paper is to demonstrate the potential for model constraint using multiple observations. As we argue in the paper (and in our replies to the reviewer comments) a global analysis would essentially be a scaled up version of what we are doing here – i.e., constraint of global ERF will be dependent on the extent to which we can constrain the model in all the key regions. Global forcing is the sum of regional forcings, and each region has its own unique combination of uncertainties.

**References:**

Boucher, O., Randall, D., Artaxo, P., Bretherton, C., Feingold, G., Forster, P., Kerminen, V.-M., Kondo, Y., Liao, H., Lohmann, U., Rasch, P., Satheesh, S. K., Sherwood, S., Stevens, B. and Zhang, X. Y.: Clouds and Aerosols, in Climate Change 2013: The Physical Science Basis. Contribution of Working Group I to the Fifth Assessment Report of the Intergovernmental Panel on Climate Change, edited by V. B. and P. M. M. Stocker, T.F., D. Qin, G.-K. Plattner, M. Tignor, S.K. Allen, J. Boschung, A. Nauels, Y. Xia, Cambridge University Press, Cambridge, United Kingdom and New York, NY, USA. 571., 2013.

Mann, G. W., et al.: Intercomparison and evaluation of global aerosol microphysical properties among AeroCom models of a range of complexity, Atmos. Chem. Phys., 14(9), 4679–4713, doi:10.5194/acp-14-4679-2014, 2014.

Regayre, L. A., Johnson, J. S., Yoshioka, M., Pringle, K. J., Sexton, D. M. H., Booth, B. B. B., Lee, L. A., Bellouin, N., and Carslaw, K. S.: Aerosol and physical atmosphere model parameters are both important sources of uncertainty in aerosol ERF, Atmos. Chem. Phys., 18, 9975-10006, doi:10.5194/acp-18-9975-2018, 2018.

Reddington, C. L., et al.: The Global Aerosol Synthesis and Science Project (GASSP): Measurements and Modeling to Reduce Uncertainty, Bull. Am. Meteorol. Soc., 98(9), 1857–1877, doi:10.1175/BAMS-D-15-00317.1, 2017.

---

## Author Comment (AC3) · 6 Aug 2018

In our response, referee comments are marked in bold, our responses and original text in plain text, and altered text in the paper in bold italic.

**Response to reviewer 3 (Anonymous reviewer)**

We thank the reviewer for their interesting and useful comments on our manuscript. Our responses to these comments are given below.

**Main comment: One comment I have, is that it should be made more clear in abstract and conclusion, and also some figures and tables, that they use synthetic observations and not real observations. And define synthetic observations the first time it is mentioned.**

We have adapted the text in the abstract and conclusions, and in the captions of Table 2 and Figures 3 and 7 to make this clearer throughout the manuscript.

– In the abstract, we have revised the following sentence on page 1 line 18:
"The model uncertainty is calculated by using a perturbed parameter ensemble that samples twenty-seven uncertainties in both the aerosol model and the physical climate model***, and we use synthetic observations generated from the model itself to determine the potential of each observational type to constrain this uncertainty.***"

– The caption of Table 2 has been adjusted to:
"**Table 2.** Observed quantities and corresponding uncertainty ranges used for the constraints applied over Europe. Values are a European July mean***, synthetically generated from the model output of a selected PPE member.***"

– The caption of Figure 3 has been adjusted to:
"**Figure 3.** Calculated uncertainty in the aerosol quantities and aerosol ERF terms from the 4 million member sample. Results are for July-mean over Europe. The red bar shows the assumed range of each ***synthetic*** observation used to constrain the uncertain parameter space and the aerosol forcing uncertainty from Table 2."

– The caption of Figure 7 has been adjusted to:
"**Figure 1.** The relative constraint achieved for aerosol ERF, ERF$_{ACI}$, ERF$_{ARI}$ and ERF$_{ARIclr}$ over Europe given the individual ***synthetic*** constraints applied (colours) and the simultaneous constraint (ALL). The relative constraint is evaluated as the ratio of the standard deviation of the forcing in the constrained sample ($\sigma_{constrained}$) to the standard deviation of the forcing in the original, unconstrained sample ($\sigma_{full}$)."

– In the conclusions section we have added the following sentence to page 28 line 17 of the original manuscript (page 30, line 16 in the revised manuscript).
"***Using synthetic observations (taken from the output of one of our simulations) we determine the extent of the potential constraint that these nine aerosol and cloud-related properties can generate.***"

Finally, we have defined the term "synthetic observations" at the point it is first mentioned in the body of the manuscript, in the introduction section on page 5 line 6 of the original manuscript (page 5 line 8 in the revised manuscript). The revised text is as follows:

"Although large observational datasets of aerosol in-situ microphysical and chemical properties are available (Reddington et al., 2017), we use synthetic observations here *– **i.e., observations that are***

***generated from a model simulation** –* to postpone addressing some of the challenges faced when comparing model output and in-situ observations (Schutgens et al., 2016a, 2016b)."

**Minor Comment 1: Page 7 line 10. Specify that it is biomass burning emissions.**

Page 7, line 10 of the original manuscript is an empty line break between paragraphs. However, we think this is referring to page 7 lines 19-20 (page 7 line 23 of the revised manuscript), and have updated the text here as follows:

"Carbonaceous **biomass burning** aerosol emissions for recent decades were prescribed using a ten year average of 2002 to 2011 monthly mean data"

**Minor Comment 2: Table 2: Indicate that this is not real observations. Useful to define Europe also. In addition to the synthetic observations, real observations are used for ToA flux, am I right?**

We have adjusted the caption for Table 2 to be clear that the observations are not real observations. (See bullet point 3 in our reply to the Main Comment above.)

We have updated the text at the end of the introduction section (page 5, line 9 of the original manuscript; page 5 line 11 of the revised manuscript) to more clearly define the Europe region that we have used. Revised text:

"The analysis is restricted to **the region of** Europe **(defined in this study by the longitude range: 12°W to 41°E, and latitude range: 37.5°N to 71.5°N)** for the month of July."

All observations used in this study, including the ToA Flux observation, are synthetic and come from a model run with all parameters set to their median value from the parameter's distribution that was obtained through our expert elicitation exercise. However, information from real observations (where available) was used to determine appropriate uncertainty ranges on the synthetic observations. For ToA flux, the uncertainty range was estimated to be in line with information from the Clouds and the Earth's Radiant Energy System (CERES) and IPCC uncertainty estimates (Hartmann et al., 2013). The information on the origin of the ToA flux observation used in this study in paragraph 2 of Section 2.6 was incorrect in our original manuscript, and we have amended paragraphs 2 and 3 of Section 2.6 to address this. The revised paragraphs are as follows:

"We use synthetic observations (Table 2) of European July-mean cloud condensation nuclei (CCN) concentration at 0.2% supersaturation at approximate cloud-base height, surface concentrations of PM2.5, mass concentrations of sulphate, OC and BC at the surface, **the outgoing shortwave radiative flux at the top of the atmosphere (ToA flux),** AOD at a wavelength of 550 nm, and the change in AOD (ΔAOD) and surface solar radiation (ΔSSR) between 1978 and 2008. The period 1978 to 2008 was originally chosen because it is an interesting period for global and regional forcing changes. Although AOD measurements are not available back to 1978, this is not vital to the present study which aims to assess potential constraint over a period with substantial aerosol changes.

The observation uncertainties are based on our judgement about the combined effect of instrument uncertainties and the uncertainty associated with measurement representativeness (colocation of high-frequency point measurements within low-spatial-resolution, monthly-mean model output subject to meteorological variability (Reddington et al., 2017; Schutgens et al., 2016a, 2016b). **Where**

*available, we have used sets of real observations to inform these judgements and estimates. For example, we selected our uncertainty range on the ToA Flux such that it is in line with information from the Clouds and the Earth's Radiant Energy System (CERES) and IPCC uncertainty estimates (Hartmann et al., 2013).* In the constraint process we also account for the emulator error (i.e., the estimated uncertainty in each of the 4 million points associated with using the emulator instead of the model itself)."

**Minor Comment 3: Figure 9: Does the color shading mean anything? Include a colorbar or remove the shading.**

The colour represents the marginal normalised sampling density (normalised across parameters) of each input parameter over its range. The parts of the marginal parameter space that are effectively ruled out by the constraint are shown in white (normalised sampling density <0.02).

The plot has been updated and a colour bar has been added. We have also updated the caption of the figure to clearly state the meaning of the colour-scale. The new figure is below.

[Figure]

**Figure 2.** One-dimensional projection of the remaining parameter space after simultaneous constraint of all atmospheric quantities and decadal trends. *The colour-scale shows the marginal normalised sampling density (normalised across parameters) of each input parameter over its range. Parts of the marginal parameter space that are effectively ruled out are shown in white (normalised sampling density <0.02).*

We have also updated a sentence on page 24 line 28 of the original manuscript (page 26 line 28 of the revised manuscript) for clarity. This sentence now reads:

"Figure 9 identifies parts of the *marginal* parameter space that are *effectively* ruled out *in white*."